# Neuronutrition and Nrf2 Brain Resilience Signaling: Epigenomics and Metabolomics for Personalized Medicine in Nervous System Disorders from Bench to Clinic

**DOI:** 10.3390/ijms26199391

**Published:** 2025-09-25

**Authors:** Maria Concetta Scuto, Carmelina Daniela Anfuso, Cinzia Lombardo, Eleonora Di Fatta, Raffaele Ferri, Nicolò Musso, Giulia Zerbo, Morena Terrana, Miroslava Majzúnová, Gabriella Lupo, Angela Trovato Salinaro

**Affiliations:** 1Department of Medicine and Surgery, Kore University of Enna, 94100 Enna, Italy; mariaconcetta.scuto@unikore.it (M.C.S.); nicolo.musso@unikore.it (N.M.); 2Department of Biomedical and Biotechnological Sciences, School of Medicine, University of Catania, 95123 Catania, Italy; daniela.anfuso@unict.it (C.D.A.); cinzia.lombardo@unict.it (C.L.); 3OASI Research Institute—Istituto di Ricovero e Cura a Carattere Scientifico (IRCCS), 94018 Troina, Italy; edifatta@oasi.en.it (E.D.F.); rferri@oasi.en.it (R.F.); 4Department of Chemical Sciences, University of Catania, 95125 Catania, Italy; giulia.zerbo@unict.it; 5Biomedical Science and Technologies and Nanobiotechnology Lab, RCCS Istituto Ortopedico Rizzoli, 40136 Bologna, Italy; morena.terrana@ior.it; 6Department of Animal Physiology and Ethology, Faculty of Natural Sciences, Comenius University, Ilkovicova 6, 84215 Bratislava, Slovakia; miroslava.majzunova@uniba.sk; 7Institute of Normal and Pathological Physiology, Centre of Experimental Medicine, Slovak Academy of Sciences, Sienkiewiczova 1, 81371 Bratislava, Slovakia

**Keywords:** neuronutrition, neuroepigenetic regulation, brain resilience, polyphenols, Nrf2 pathway, autism, nervous system disorders, metabolomics

## Abstract

Neuronutrition to improve brain resilience to stress and human health has received considerable attention. The use of specific nutrients is effective in preventing and slowing neurodegenerative and neuropsychiatric disorders. Selective neuronutrients, including polyphenols, short-chain fatty acids (SCFAs), tryptophan, tyrosine, and sulfur metabolites, can modulate the dysregulated nuclear factor erythroid 2 (Nrf2) pathway through neuroepigenetic modifications and altered levels of neurotransmitters such as serotonin, melatonin, and dopamine. In particular, abnormal epigenetic alterations in the promoter function of the NFE2L2/Nrf2 gene may contribute to the onset and progression of various diseases by disrupting cellular homeostasis. Recent evidence has documented that polyphenols are capable of modulating Nrf2 signaling; to do this, they must reverse hypermethylation in the CpG islands of the NFE2L2 gene. This process is achieved by modifying the activity of DNA methyltransferases (DNMTs) and histone deacetylases (HDACs). Furthermore, a diverse group of polyphenolic metabolites can be identified and quantified using innovative mass spectrometry platforms in both in vitro models and human urine samples to investigate redox metabolic homeostasis under physiological and pathophysiological conditions. This review aims to deepen the current understanding of the role of nutrient-derived secondary metabolites. It highlights innovative strategies to effectively prevent, slow, or potentially reverse neuroinflammation and oxidative stress, key drivers of neuronal damage. The targeted application of these metabolites can be considered a novel, personalized neuronutritional approach to promote brain health and neuronal adaptation.

## 1. Introduction

Neuronutrition, recently proposed as part of nutritional neuroscience, represents a valid model of personalized nutritional medicine that promotes preventive, therapeutic, and neurorehabilitative effects of various food nutrients on cognitive function and behavioral disorders. Supplementation with neuronutrients, particularly functional polyphenols, has proved to be an appropriate therapeutic strategy for preventing or delaying brain disorders, especially Alzheimer’s disease (AD), Parkinson’s disease (PD), and autism spectrum disorders (ASD), due to their neuroprotective effects, availability, and safety [1]. Neurodegenerative diseases (NDs) and neuropsychiatric diseases are complex due to their intricate pathophysiology and the lack of available treatments. Neuropathological features include neuronal loss and functional impairment, with significant global health impacts. Currently, there is no effective and definitive remedies for AD, PD, and ASD. Emerging evidence suggests that neuronutrients, such as polyphenols, promote brain resilience and neuroprotective effects by improving cognitive function and mitigating neuroinflammation, oxidative damage, and neuronal death in cellular and animal models and in humans [2,3,4]. Recent nutritional research has demonstrated that specific nutrients may have therapeutic potential for neuropathological conditions such as anxiety, depression, and degenerative and neurodevelopmental disorders [5,6,7]; these findings often remain theoretical with limited clinical application. Recent evidence indicates that neuronutrients and eating behavior can influence the pathogenesis of various nervous system disorders, as well as patients’ cognitive and emotional states [8]. Other researchers have defined neuronutrition as not only the use of diet but also the use of various nutrients to prevent and manage central and peripheral nervous system disorders across the lifespan [9]. Interestingly, neuronutrients can influence AD and PD development [10,11] and offspring behavioral impairment [12]. In a broader concept, neuronutrition is an interdisciplinary field that examines how nutrition influences brain health and cognition [13,14]. It studies the impact of various elements, such as nutrients, diet, eating behavior, the nutritional environment, nutriepigenomics, and metabolomics, on maintaining the brain and its functions in optimal condition. In the context of neurodegeneration such as AD and PD, the cumulative effects of oxidative stress and inflammation throughout an individual’s lifetime are among the main factors contributing to disease onset and progression [15]. Several studies, primarily in NDs, have emphasized the promising potential of various neuronutrients and dietary habits in stemming or at least limiting stress and neuroinflammation, as well as strengthening antioxidant defenses and brain resilience pathways [16,17,18,19,20]. Similarly, polyphenol metabolites dose-dependently influence neurobehavioral and physiological responses by targeting the Sirt1 pathway to inhibit oxidative damage and inflammatory mediators in multiple models of neuropsychiatric disease [21,22]. Oxidative stress plays a substantial role in neuroinflammatory and ND diseases by causing cellular damage and mitochondrial dysfunction, which contribute to aging and the progression of neuronal loss. The brain has a remarkable capacity to adapt. When mild neuronal stress occurs, the brain activates resilience mechanisms that protect brain cells and maintain optimal cognitive function. This has a protective effect. However, if stress exceeds the brain’s defense capacity, neurons activate defense pathways to protect themselves, blocking the excessive production of reactive oxygen species (ROS), substances that damage cells. This review aims to examine current knowledge of the neuronutrition approach and its fundamental role in brain health, with the goal of identifying novel candidate biomarkers and perspectives for future medicine through personalized nutritional therapy that may effectively prevent or attenuate oxidative stress, neuroinflammation, and metabolic imbalances, which can trigger neurodegeneration and behavioral alterations. Moreover, we provide an overview of the main polyphenol metabolites targeting the Nrf2 pathway to stimulate a brain resilience response in order to prevent and/or treat major nervous system disorders, based on metabolomic approaches and neuroepigenetic interactions.

## 2. Neuronutrition Targeting Nrf2 Brain Resilience Signaling in Nervous System Disorders: Personalized Preventive and Therapeutic Innovations

These neurosensitive pathways involve the activation of Nrf2/NFE2L2, a key regulator gene of the antioxidant response element (ARE) pathway. Specifically, Nrf2 translocates from the cytosol into the nucleus and binds to ARE in the promoter regions orchestrating the expression of various brain resilience target genes and enzymes such as heme oxygenase-1 (HO-1), heat shock protein 70 (Hsp70), the thioredoxin (Trx)/thioredoxin reductase system, sirtuin-1 (Sirt1), NADPH quinone oxidoreductase 1 (NQO1), γ-glutamylcysteine synthetase (γ-GCS), and Forkhead box O (FoxO) involved in detoxification, redox balance, and cellular defense against oxidative stress. For instance, FoxO is involved in the glutathione redox system and helps prevent or block neuropathological features (e.g., oxidative stress and neuroinflammation) that contribute to brain dysfunction (maladaptive stress responses) and the formation or progression of chronic CNS disorders [23,24,25].

Neuronutrition is an emerging field of neuroscience that evaluates the effects of food components and/or supplements, especially flavonoids such as ellagic acid, anthocyanins, caffeoylquinic acid, genistein, fisetin (Figure 1), and flavan-3-ols, as well as SCFAs, tyrosine, and tryptophan metabolites, on the central nervous system, behavior, cognitive function, brain resilience, and overall health (Figure 2) [26]. In the context of neuronutrition for brain health, the primary objectives are as follows: (1) reducing damage caused by free radicals; (2) modulating the inflammatory response in the nervous system; (3) improving mitochondrial function; and (4) restoring the proper balance of neurotransmitters, the molecules that enable communication between neurons [10]. Brain resilience is a complex neuronal response that improves psychological and behavioral adaptability to life adversities, such as trauma or intense stress. This occurs through several neuroadaptive mechanisms, including the hypothalamic–pituitary–adrenal (HPA) axis, the autonomic nervous system (ANS), brain-derived neurotrophic factor (BDNF), neuroepigenetic changes, anti-apoptotic and anti-inflammatory pathways, and Nrf2 redox resilience pathways. This intrinsic capacity allows individuals to cope with or avoid stressful conditions or disorders, challenging the idea that extraordinary abilities are required for stress adaptation [1,3,24].

The evolution of the “food as healing medicine” paradigm reflects a broader shift in nutritional science toward proactive, health-oriented dietary strategies. Consistent with this, food is not only a source of energy and essential nutrients but also a potential vehicle for disease prevention and therapeutic intervention [1]. From this new perspective, adopting a personalized neuronutritional approach can identify specific epigenetic profiles and brain resilience targets, thus helping to restore the impaired Nrf2 signaling pathway [3].

Recent research integrates epigenomic and metabolomic data to understand how genes influence individual nutritional needs and disease susceptibility [27]. This will allow us to characterize vulnerable individuals to biologically resilient individuals to environmental challenges, such as centenarians, who serve as models of longevity and healthy aging [27]. Neuronutrients, including ellagic acid, anthocyanins, caffeoylquinic acid, fisetin, and genistein protect neurons against injury and disease in a dose-dependent manner. Recent preclinical and clinical studies observed that a sub-toxic dose of food nutrients such as flavonoids and their metabolites can trigger adaptive neuronal stress responses driven by the modulation of resilience genes [1,3,24]. For this purpose, activating Nrf2 pathway and brain resilience phase II genes, neuronutrients may potentially prevent or mitigate environmental challenges, namely oxidative stress and neuroinflammation, thus preserving brain health and counteracting the initiation and progression of NDs and neuropsychiatric diseases [25]. Therefore, we propose this neuronutritional approach as part of personalized medicine to apply preventive and therapeutic interventions in neuropathological conditions, mainly concerning AD, PD, and autism. All these pathologies are rooted in altered neuroadaptation to stress [28].

### 2.1. Ellagic Acid

Ellagic acid (2,3,7,8-tetrahydroxybenzopyrano [5,4,3-cde]benzopyran-5,10-dione; EA) is a flavonoid naturally present in berries and pomegranates, known for its antioxidant, anti-inflammatory, and neuroprotective properties [29]. The intestinal bacterial flora converts this substance into a class of metabolites known as urolithins (urolithin A–D). It is important to note that EA, along with some of its metabolites (especially urolithin A), can block various oxidant and inflammatory signaling pathways. By acting on mechanisms such as Nrf2, mitogen-activated protein (MAPK), and JAK-STAT, these substances help reduce chronic inflammation that accelerates the aging process [30]. The recent literature reports that EA supplementation, at appropriate doses, exhibits preventive and therapeutic potential against ND disorders and ASD via the induction of the Nrf2 pathway and brain resilience phase II genes, Table 1 [31].

#### 2.1.1. Brain Resilience Potential of Ellagic Acid in AD

Preclinical research has shown that EA at a concentration of 60 μM stimulates the production of GPX4 (an antioxidant protein). This process activates the Nrf2/Keap1 signaling axis, which in turn promotes the elimination of free iron. Consequently, EA reduces oxidative stress, ferroptosis (a type of cell death), and cognitive dysfunction caused by arsenic in hippocampal neuronal cells [32]. Furthermore, EA at concentrations of 10 and 25 μM downregulates proinflammatory cytokines such as IL-1β and upregulates the Nrf2 pathway and stress resilience proteins, including HO-1, NQO1, and SOD, under pro-oxidant conditions in vitro [33]. Moreover, EA (33.1 μM), and in particular its metabolite urolithin A (13.1 μM), has the potential to modulate the aging clock by stabilizing Sirt1 protein, which likely enhances Bmal1 oscillation in senescent human fibroblast cells in a dose-dependent manner [34]. Experimental in vivo studies have reported the neuroprotective effects of EA (50 mg/kg) on hippocampal memory function by upregulating Nrf2 and Bcl-2 through stimulation of the cAMP response element binding (CREB) and IRS/PI3K/Akt/GS3Kβ axis in rats [35]. Interestingly, EA is a substance capable of reducing the damage caused by sleep deprivation by activating the Nrf2/HO-1 pathway, the defense system that helps protect cells from oxidative stress, and attenuating the TLR4-induced inflammatory response. Thus, in this way, EA prevents memory impairment and reduces anxiety in rodents [36]. Oral administration of EA at 100 mg/kg attenuates post-surgical cognitive decline and hippocampal oxidative stress by increasing the activity of the Nrf2/HO-1 pathway and brain resilience enzymes such as SOD and CAT, targeting the insulin-like growth factor-1 (IGF-1) pathway in aged mice [37]. Finally, a clinical study on 18 patients with high oxidative stress levels who received three capsules of Robuvit^®^ (Horphag Research Ltd., Hoboken, NJ, USA) supplementation at a dose of 100 mg per day, for 8 weeks (containing EA and urolithins), reported significant improvements in fatigue and insomnia (Figure 1) [38].

#### 2.1.2. Brain Resilience Potential of Ellagic Acid in PD

Emerging preclinical studies have shown that EA supplementation at doses ranging from 5 to 20 µM effectively attenuates cadmium-induced cytotoxicity in hippocampal neuronal HT22 cells and reduces the ratio of Bax/Bcl-2 and cleaved caspase-3 protein expression by inhibiting oxidative stress and apoptosis via upregulation of the Nrf2/HO-1 signaling pathway [39]. Likewise, administration of urolithin A significantly reduced 6-OHDA-induced mitochondrial dysfunction and neurotoxicity in PC12 cells, which was accompanied by increased mitochondrial biogenesis via activation of the SIRT1–PGC-1α signaling pathway [40]. Moreover, EA (25–100 µM) significantly inhibited α-synuclein-induced cytotoxicity and aggregation by enhancing anti-apoptotic Bcl-2 and reducing serine/threonine kinase (AKT) levels via autophagy mechanisms in human SH-SY5Y neuroblastoma cells as a model of PD, in a dose-dependent manner [41]. In addition, the same authors reported that a dose of 10 mg/kg EA treatment before MPTP injection effectively restored dopaminergic neuron loss by enhancing the levels of antioxidant enzymes, preventing GSH depletion, and repressing lipid peroxidation and pro-inflammatory cytokines [42]. Recent data from Ardah and coworkers suggested that EA (100 mg/kg) exerts neuroprotective effects on dopaminergic neurons through the activation of astroglial Nrf2 signaling in PD rodent models [43]. Administration of a moderate dose (10 mg/kg) of EA reduced α-synuclein spreading and preserved dopaminergic neurons in a male C57BL/6 mouse model of PD [44]. In an experiment on rat models of Parkinson’s disease, a dose of 50 mg/kg/day of EA administered for one week in the striatum of the brain reduced damage, showing significant positive effects: it decreased levels of malondialdehyde, reactive oxygen species, and DNA fragmentation, all indicators of cellular damage. Furthermore, there was an improvement in enzymatic activity, such as monoamine oxidase B, an enzyme involved in the degradation of neurotransmitters, which is often overactive in Parkinson’s. These benefits were achieved by activating the ERβ/Nrf2/HO-1 signaling cascade, a pathway that protects cells from oxidative stress in a 6-OHDA rat model of PD [45]. Also, EA (50 mg/kg) ameliorated LPS-induced DA neuronal loss and toxicity and inhibited microglial NLRP3 inflammasome signaling activation in the substantia nigra of rats (Figure 1) [46].

#### 2.1.3. Brain Resilience Potential of Ellagic Acid in Autism

Currently, there are few studies on the neuroprotective role of EA in autism. Among these, an interesting report found that ethanol exposure impaired cognitive and mitochondrial functions, causing oxidative stress and inflammation in the brain. However, EA (20 and 40 mg/kg) administration effectively prevented the toxic effects of ethanol in the fetal alcohol spectrum disorders model [47]. Overall, EA and its active metabolites can be considered as an innovative personalized neuronutritional approach in the prevention and management of nervous system disorders to potentially enhance brain resilience and the overall quality of life in humans (Figure 1).

### 2.2. Anthocyanins

Anthocyanins and proanthocyanidins are the major bioactive flavonoids found in grapes and blueberries [48]. Anthocyanin metabolites are promising therapeutic neuronutrients for the prevention of nervous system disorders and the promotion of brain resilience, Table 1 [49]. Notably, protocatechuic acid exhibited improvements of the cognition function and memory, a reduction in brain contents of MDA, and an elevation of GSH and SIRT1 [49].

#### 2.2.1. Brain Resilience Potential of Anthocyanins in AD

Preclinical evidence has shown that a dietary anthocyanin, cyanidin-3-O-glucoside (50 μM), reduced inflammatory cytokines and induced a shift in microglial phenotype from M1 to M2 by downregulating M1-specific markers (CD80 and CD86) through activation of PPARγ and the triggering receptor expressed on the myeloid cells 2 (TREM2) pathway, enhancing amyloid beta (Aβ)42 phagocytosis in HMC3 cells [50]. Similarly, administration of 30 mg/kg/day cyanidin-3-O-glucoside for 38 weeks decreased inflammatory cytokines and increased expression of M2-specific markers (CD206 and Arg1) in an APPswe/PS1ΔE9 rodent model [50]. A study demonstrated that anthocyanins reduce oxidative stress and ROS formation caused by protein aggregates (AβO), activating the PI3K/AKT/Nrf2 signaling pathway. Furthermore, these substances prevent programmed cell death and neuronal degeneration by suppressing key markers such as caspase-3 and PARP-1 in hippocampal HT22 cells and animal models of Alzheimer’s disease (APP/PS1 mice) [51]. A recent randomized, double-blind clinical study has highlighted positive effects: the intake of 80 mg capsules of naturally purified anthocyanins, for 24 weeks, significantly improved cognitive functions in elderly subjects at high risk of dementia, compared to those who had received a placebo [52]. Furthermore, blueberry supplementation showed improvements in long-term memory performance in older adults with MCI [53] and executive function ability in healthy older adults [54,55]. More recently, a randomized double-blind, placebo-controlled study conducted by Krikorian and coworkers demonstrated that the dose of 0.5 c of whole fruit of blueberry supplementation had neurocognitive benefits in middle-aged individuals with insulin resistance and elevated risk for future dementia by selectively improving executive function and memory deficits after 12 weeks [56].

#### 2.2.2. Brain Resilience Potential of Anthocyanins in PD

Few studies have examined the neuroprotective efficacy of anthocyanin treatment modalities against PD. However, a recent study observed that anthocyanin-rich extracts alleviated paraquat-induced dopaminergic cell death and neurotoxicity in midbrain cultures by activating the Nrf2 pathway and ameliorating mitochondrial deficits [57]. Interestingly, polyphenol-rich blueberry juice was more effective against 6-hydroxydopamine (6-OHDA) toxicity than either treatment alone, improving behavioral performance and reducing the loss of striatal dopamine terminals concomitant with an increase in nigral glial cell-line derived neurotrophic factor (GDNF) expression in vivo [58]. In Caenorhabditis elegans, blueberry extracts (100, 200 and 400 μg/mL) dose-dependently attenuated α-synuclein protein expression, improved healthspan in the form of motility, and restored lipid content mediated by activation of sir-2.1 levels (ortholog of mammalian Sirt-1) [59]. Finally, a pilot study conducted by Fan et al. reported that the intake of blackcurrant anthocyanins capsules (300 mg) taken twice daily enhanced the concentration of cyclic glycine-proline (cGP), a metabolite of IGF-1 in the cerebrospinal fluid and the cGP/IGF-1 ratio in plasma of PD patients after four weeks [60].

#### 2.2.3. Brain Resilience Potential of Anthocyanins in Autism

Recently, evidence has shown the neuroprotective and therapeutic effects of anthocyanins in alleviating autism-like symptoms [61]. Notably, an anthocyanin-rich Portuguese blueberry extract showed significant positive effects in the offspring of mice exposed to valproic acid (VPA) prenatally. Administration of a dose of 30 mg/kg/day resulted in a significant reduction in oxidative and inflammatory markers in both the brain and gut. At the same time, serotonin levels were increased in the prefrontal cortex of the brain and the gut. These changes resulted in improved social behaviors and a decrease in repetitive behaviors in this mouse model [61]. Currently, there are few studies on brain resilience and the potential therapeutic effects of anthocyanins in autism, and no clinical trials exist. There is a need to study neuro-nutraceutical compounds in this promising but still underexplored field.

### 2.3. Centella asiatica and Caffeoylquinic Acid Metabolites

Caffeoylquinic acids (CQAs), esters of caffeic acid with quinic acid (Figure 1), are secondary metabolites found in edible and medicinal plants from various families, especially *Centella asiatica* and many others. Recent evidence has demonstrated that CQAs have antibacterial, antioxidant, anti-inflammatory, and neuroprotective activities [62,63,64]. Importantly, CQAs exhibit the pharmacological potential to prevent or slow NDs and psychiatric diseases by activating the Nrf2 pathway and related brain resilience genes in experimental models and humans, Table 1 [65,66,67].

#### 2.3.1. Brain Resilience Potential of Caffeoylquinic Acids in AD

In AD pathogenesis, CQAs (25 μg/mL) from Lonicera japonica Thunb alleviated inflammation induced by LPS plus IFN-γ by directly downregulating transforming growth factor-β-activated kinase 1 (TAK1), c-jun N-terminal kinase (JNK), and c-JUN and upregulating Nrf2/HO-1 signaling under exogenous stress conditions [65]. Interestingly, Jiang and colleagues have shown that caffeoylquinic acid derivatives from burdock roots at doses ranging from 7.5 to 30 μM exerted cellular neuroprotection against H_2_O_2_ damage by reducing oxidative stress and the phosphorylation of MAPK signaling pathways including extracellular signal-regulated kinase 1/2 (ERK1/2), JNK, and p38 and increasing intracellular GSH-Px and SOD levels and phosphorylation of AKT in a dose-dependent manner [66]. Furthermore, mono and dicaffeoylquinic acids found in *Centella asiatica* improved cognitive performance by attenuating amyloid β-toxicity in both neuroblastoma MC65 cells [68] and in 5XFAD AD rodent models [69]. Administration of caffeic acid phenethyl ester (at a dose of 10 mg/kg) to the brains of mice has been shown to counteract the damaging effects of amyloid β oligomers. The treatment activated the Nrf2 and HO-1 defense pathways by modulating the enzyme glycogen synthase kinase 3β in the hippocampus. Additionally, the same treatment reduced inflammation and neuronal cell death, while improving spatial learning and recognition memory in mice [70]. A double-blind, randomized clinical study observed the oral bioavailability and pharmacokinetics of *Centella asiatica* water extract product (CAP) (2 g and 4 g doses) in four mildly demented older adults. Notably, in these patients, a low concentration of 2 g of CAP revealed a potent dose–response effect, ameliorating cognitive decline related to increased abundance of its phase II metabolites including triterpene aglycones, asiatic acid, madecassic acid, and mono- and di-CQA in both plasma and urine, mainly targeting the Nrf2 pathway and related genes [67]. More recently, the same authors validated a method for the isolation and detection of CQAs from *Centella asiatica* in human urine and plasma using HPLC-MS/MS technology. The study observed that oral absorption of triterpenes and CQAs from CAP can potentially prevent the risk of dementia in healthy older adults in a dose-dependent relationship (Figure 1) [71].

#### 2.3.2. Brain Resilience Potential of Caffeoylquinic Acids in PD

Recently, studies showed that the phenolic compound of Corema album berry juice, 5-O-caffeoylquinic acid, induced dose-dependent neuroprotection of dopaminergic and cholinergic cells exposed to 6-OHDA and okadaic acid by effectively inhibiting the activity of monoamine oxidase A (MAO-A) and B (MAO-B) enzymes in vitro [72]. A walnut leaf extract, at doses of 1.7 and 17 mg/mL, along with its active metabolites (trans-3-caffeoylquinic acid and quercetin-3-hexoside), showed significant positive effects. These compounds significantly increased resistance to thermal and oxidative stress, also improving fertility and offering protection to the nervous system by delaying Aβ-induced paralysis and accumulation of α-synuclein aggregates in *C. elegans* transgenic models of AD and PD (Figure 1) [73].

#### 2.3.3. Brain Resilience Potential of Caffeoylquinic Acids in Autism

*Centella asiatica* and its active metabolites are considered neuroregenerative herbal medicines used to attenuate memory deterioration and repetitive and stereotyped behaviors [74]. A recent study reported that ethanolic extract of *Centella asiatica* given orally at doses of 150 and 300 mg/kg b.w./day significantly attenuated AlCl3-induced neurotoxicity and behavioral alterations in the rat brain after 60 days. Indeed, co-administration of ethanolic extract of *Centella asiatica* protected the brain from AlCl3-induced cognitive dysfunction and oxidative stress, through an activation of SOD and catalase activity in the hippocampus of rats (Figure 1) [75].

### 2.4. Genistein

Genistein (GEN), a secondary metabolite that belongs to a family of polyphenolic compounds called isoflavones, is present in soy and has been shown to have numerous pharmacological properties, particularly against CNS disorders in a dose-dependent manner, Table 1 [76,77]. Interestingly, genistein-7-glucuronide (G-7-G), genistein-7-sulfate (G-7-S), and 4′-sulfate (G-4′-S) metabolites were detected and quantified in human plasma of 12 healthy young volunteers [78].

#### 2.4.1. Brain Resilience Potential of Genistein in AD

The brain resilience ability of GEN in modulating multiple brain resilience signaling pathways to prevent or attenuate Aβ-induced toxicity, neuronal apoptosis, memory impairment, and imbalance of neurotransmitter receptor levels, as well as promoting antioxidant effects, has been demonstrated in vitro and in vivo [79,80,81,82,83,84,85,86]. Consistent with the relevant notion, studies revealed that GEN can modulate Nrf2/HO-1/PI3K signaling to treat AD pathogenesis in neuroblastoma SH-SY5Y cells [80]. Likewise, GEN (0.1–1 μg/mL) effectively blocked the decreases in α7 nicotinic acetylcholine receptor mRNA and protein expression in primary hippocampal neurons treated with Aβ25-35 mediated via the activation of PI3K/Akt/Nrf2 signaling [81]. Interestingly, GEN-loaded chitosan nanoparticles at doses ranging from 0.5 to 2.0 mg exhibited no cytotoxic or apoptotic effects on pheochromocytoma (PC12) cells. Instead, in ex vivo experiments, a dose of 50 µg of GEN showed the potential to penetrate through the nasal mucosa and reach the brain for neuroprotection as compared to pure GEN [82]. Moreover, GEN alleviated scopolamine-induced amnesia in mice by boosting cholinergic neurotransmission, improving the antioxidant system, and activating the ERK/CREB/BDNF signaling [83]. Other studies showed that GEN (10 mg) administered orally for 10 days significantly attenuated cognitive impairment and synaptotoxicity in the hippocampus by downregulating GSK-3β/ERK/JNK pathways in a rat model of Aβ peptide-induced toxicity [84]. Another study reported that the GEN could successfully enhance learning and memory capacity, lower hippocampal neuron deterioration, and decrease GRP78, CHOP, Caspase-12, Cle-Caspase-9, Cle-Caspase-3, PERK, and p-PERK protein expression levels [86]. Recent in silico studies using an artificial intelligence (AI) network have shown that GEN and quercetin present in extra virgin oil induced neuroprotective effects on cognitive deficits and AD management [87]. Importantly, a low dose of GEN administration (15 mg) improved memory and cognitive dysfunction through a marked reduction in oxidative stress parameters (MDA and protein carbonyls) and the enhancement of the Nrf2 pathway (e.g., HO-1 and NQO-1) and the antioxidant enzymes (e.g., SOD and CAT) in the hippocampus of septic rats [88]. Emerging clinical studies reported that one capsule of GEN (60 mg) or a placebo, administered orally twice per day for up to 12 months, delays the onset of AD in patients with mild cognitive impairment [89]. More recently, the same authors demonstrated that GEN significantly improved cognitive preservation among individuals with prodromal AD [90]. Finally, a pilot study on 46 patients that received 50 mg of daily supplementation of PhytoSERM (a formulation of three selective isoflavones—genistein, daidzein, and S-equol) or placebo showed significantly greater reduction in menopause-associated hot flash frequency, ultimately preserving cognitive decline through improved verbal learning and executive function (Figure 1) [91].

#### 2.4.2. Brain Resilience Potential of Genistein in PD

Preclinical research indicates that the GEN isoflavone may be effective for treating PD as reported by in vitro and in vivo evidence [92,93,94,95]. In this regard, Liu and colleagues revealed that the phytoestrogen GEN, like estrogen, exhibits neuroprotective effects against 1-methyl-4-phenyl-1,2,3,6-tetrahydropyridine (MPTP)-induced neuronal injury in the nigrostriatal system by enhancing Bcl-2 gene expression in dopaminergic neurons [92]. In human SH-SY5Y cells overexpressing A53T mutant α-synuclein, a dose of 20 μM of GEN has been shown to inhibit oxidative stress and apoptosis by activating estrogen receptors and NFE2L2 channels [93]. Importantly, Du and co-workers found that GEN decreased LPS-induced nigro-striatal damage in an ovariectomized rat model of PD, prevented microglial over-activation, and protected dopaminergic neurons via G protein–coupled estrogen receptor (GPER) and insulin-like growth factor 1 receptor (IGF-1R) signaling pathways [94]. Furthermore, GEN treatment effectively restored the impaired spatial learning and memory and prevented 6-OHDA-induced neuronal loss in PD animals (Figure 1) [95].

#### 2.4.3. Brain Resilience Potential of Genistein in Autism

Autism is a prevalent neuropsychiatric disorder characterized by repetitive behaviors and impaired social communication, and is often associated with depression and anxiety conditions [96]. Of note, GEN can modulate key neural pathways by reducing neuroinflammation, restoring neurotransmitter balance, and improving both behavioral and cognitive functions, highlighting its potential as a novel therapeutic candidate for autism [97,98]. Recent data suggested that GEN (40 and 80 mg/kg., orally) significantly improved locomotion, neuromuscular coordination, and cognitive functions, restored levels of AC, cAMP, CREB, PKA signaling molecules and mitochondrial complexes (I-V), reduced neuroinflammation (TNF-α and IL-1β) and apoptosis-related proteins (Bax, Bcl2, Caspase-3), and normalized neurotransmitter levels (acetylcholine, dopamine, GABA, and serotonin) in vitro and in vivo [97]. A multi-omic approach revealed a sex-dependent inter-relationship between GEN-induced microbiome, metabolome, and socio-communicative behaviors in autistic rodent models [98]. Specifically, perinatal GEN (250 mg/kg) exposure induced sex-dependent changes in gut microbiota and metabolites. In female mice, GEN exposure predominantly decreased several metabolites, including γ-aminobutyric acid (GABA), cysteine, homoserine, ornithine, and glycine, while it upregulated 3-amino-isobutanoic acid and down-regulated methyl linoleate, N2-acetyl-ornithine, daidzein, α-tocopherol, and 3β,5β-choesetan-3-ol, as well as reduced Bacteroidetes bacteria, which were positively associated with 1-heptadecanol and 3μ,5μ, cholestan-3-ol in male rodent models of autism [98]. Overall, preclinical findings highlight the neuroprotective role of GEN in attenuating oxidative stress, neuroinflammation, and neurotransmitter imbalance. Currently, there are no clinical data evaluating the therapeutic effects of GEN in humans (Figure 1).

### 2.5. Fisetin

Fisetin (3,3,4,7-tetrahydroxyflavone), a flavonoid molecule (Figure 1) abundant in various fruits and vegetables including strawberries, apples, onions, and persimmons, has shown anti-oxidant, anti-degenerative, anti-inflammatory, analgesic, and anti-tumor effects by targeting brain resilience pathways including Nrf2 signaling and related genes to counteract the onset of nervous system disorders, Table 1 [99,100,101]. Of note, the biologically active metabolite of fisetin, 3,4′,7-trihydroxy-3′-methoxyflavone, was identified in plasma samples of healthy subjects [102].

#### 2.5.1. Brain Resilience Potential of Fisetin in AD

Preclinical evidence has demonstrated that low/moderate doses (0.1 g/L) of fisetin increased stress resistance, extended the lifespan, and delayed age-related cognitive decline and Aβ-induced paralysis in *C. elegans* via the activation of autophagic mechanisms in a dose-dependent manner [103]. Interestingly, fisetin at doses ranging from 25 to 75 mg in a dose-dependent pattern reduced neurotoxicity and neurodegeneration by upregulating brain resilience enzymes including glutathione S-transferase (GST) levels and downregulating the release of IL-6 in adult albino rats after 21 days [104]. Most recently, Rakshit et al. reported that an oral dose of 5 mg per day of chitosan-coated fisetin nanoformulation markedly reduced oxidative stress and pro-inflammatory cytokines along the cortex, hippocampus, and colon and improved memory impairment, motor deficits, and depression-like symptoms induced by Aβ1-42 through the activation of Nrf2/HO-1 in AD mice [105]. Finally, a dose of 20 mg/kg/day for 30 days of fisetin exerted neuroprotective effects against D-galactose-induced oxidative stress-mediated neuroinflammation and memory impairment by enhancing the endogenous antioxidant Nrf2/Sirt1 signaling in mice [100] (Figure 1).

#### 2.5.2. Brain Resilience Potential of Fisetin in PD

In PD, fisetin (10 and 20 mg/kg p.o.) reversed rotenone-induced behavioral deficits, mitochondrial enzyme dysfunctions, and aberrant dopamine levels by enhancing stress resilience pathways in particular GSH and catalase levels in rats in a dose-dependent manner [106]. Additionally, studies reported that fisetin (100 ng/kg) significantly inhibited MPTP-induced dopaminergic neurodegeneration. The potential mechanism underlying the neuroprotective effect of fisetin involved the regulation of the gut microbiota, with increased abundance of Lachnospiraceae and decreased abundance of Escherichia-Shigella and Bacillus in a mouse model of PD [107]. Lastly, the nanoemulsifying drug delivery system of fisetin significantly exhibited neuro-therapeutic potential in maintaining cellular redox homeostasis by upregulating Nrf2 and brain resilience enzymes including GSH, SOD, and CAT in PD rats (Figure 1) [108].

#### 2.5.3. Brain Resilience Potential of Fisetin in Autism

Fisetin has shown promising potential for the prevention and treatment of neurodevelopmental conditions such as autism [109,110]. Accordingly, recent studies indicated that low doses of 10 mg/kg/b.w. daily gestational and post-weaning fisetin treatment significantly improved behavioral impairments by attenuating elevated oxidative stress, ROS, and lipid peroxidation, and re-establishing redox homeostasis in autistic-like neurobehavioral animals [109]. Most recently, the same authors have shown that oral administration of 10 mg/kg gestational fisetin efficiently regulated ROS overproduction and prenatal oxidative stress by restoring GSH levels and reducing BBB permeability and apoptosis via inhibition of the mitochondria–Wnt signaling axis in VPA-induced autism-associated symptoms in rats (Figure 1) [110].

**Table 1 ijms-26-09391-t001:** Effects of neuronutrients in molecular pathways involved in AD, PD, and autism.

Neuronutrients	Molecular Pathways	AD	PD	Autism	Ref.
Ellagic acid	↓ NF-κB ↓ JAK-STAT ↑ Nrf2	↓ NF-κB, IL-1β, TLR4 ↑ Nrf2/Keap1 ↑ IRS/PI3K/Akt/GS3Kβ	↓ Bax/Bcl-2, caspasi-3 ↓ 6-OHDA, Cox-2 ↑ Nrf2/HO-1	- - -	[30,33,36] [30,32,42,46] [31,35,39]
Anthocyanins	↓ NF-κB	↓ IL-Iβ, IL-6, TNF-α ↑ PI3K/AKT/Nrf2	↓ 6-OHDA	↓ COX-2 ↓ IL-1β, IL-6 ↓ TNF-α	[50,51,58] [51] [61]
Caffeoylquinic acids	↑ Nrf2	↓ JNK, c-JUN ↑ ERK1/2 ↓ AKT, GSK-3β	↓ 6-OHDA	-	[65,70,72] [66]
Genistein	↑ Nrf2/HO-1/PI3K	↑ PI3K/Akt/Nrf2 ↑ ERK/CREB/BDNF ↑ GSK-3β/ERK/JNK ↓ NF-kB, TNF-α, IL-1β	↓ 6-OHDA	↓ TNF-α, IL-1β ↓ Bax, Bcl2 ↓ Caspase-3	[80,81,95] [83,97] [84,98] [88]
Fisetin	↑ Nrf2	↓ p-JNK/NF-kB, IL-6 ↑ GST ↑ Nrf2/HO-1	↓ TNF-α, IL-6 ↑ GSH, SOD, CAT	↑ GSH	[99,100,110] [104,108] [105]

(↓ down regulation, ↑ up regulation).

## 3. Neuroinflammation: Role of Neuronutrients

Neuroinflammation is a pathological response of the secondary injury cascade implicated in most neurodegenerative processes [111], neurobehavioral alterations [112], and brain aging mechanisms [113]. This process is characterized by excessive activation of several types of cell glia, both in the peripheral system, as Schwann cells and satellite glial cells, and in the central nervous system as microglia, astrocytes, and oligodendrocytes, in areas such as the spinal cord, trigeminal nerve ganglia, central glia, and the brain [111]. Recent studies have pointed out that elevated levels of Aβ trigger microglia activation, which in turn promotes the release of pro-inflammatory cytokines and stimulates the production of amyloid precursor proteins, exacerbating neurodegenerative lesions [114]. Growing evidence indicates a tight inter-relationship between pro-oxidant stress and inflammatory response. Indeed, free radicals induce ROS that promote the release of pro-inflammatory mediators [115]. ROS overproduction by immune cells at the site of inflammation exacerbates oxidative stress and tissue damage, causing the development of chronic diseases [116]. For instance, a diet high in processed foods—i.e., products with added sugars, saturated and trans fatty acids, and sodium—can promote chronic inflammation and increase the risk of neurological diseases [117]. A large study conducted on more than 70,000 individuals highlighted a correlation between high consumption of ultra-processed foods and a higher risk of dementia [118]. The same association was also observed in young people, where a high intake of ultra-processed foods was linked to an increased incidence of depressive symptoms [119]. Indeed, increased consumption of fructose, particularly in processed foods, has been linked to mitochondrial dysfunction, increased ROS production, and neuroinflammation, all of which contribute to cognitive decline and impairments in memory and learning [119]. However, adequate intake of neuronutrients activating the expression of brain resilience proteins in microglia has the ability to control neuroinflammation and limit oxidative stress-induced neuronal damage [116,120]. As a result, personalized therapeutic interventions through neuronutrients targeting the TLR4-NF-kB pathway and related proteins, such as iNOS, COX-2, and cytokines (e.g., IL-1β, IL-2, IL-6 and TNF-α), may potentially inhibit chronic neuroinflammation and activate brain resilience signaling to mitigate neuronal death during NDs [120,121,122,123,124].

### 3.1. Neuronutrients Inhibit Neuroinflammatory Cascade and Promote Brain Resilience

Neuronutrients including plant flavonoids, i.e., flavan-3-ols, flavanones, flavones, flavanols, isoflavones, and anthocyanins, can alter multiple neuroinflammatory pathways exhibiting dose–response profiles [125]. Eating a diet rich in whole plant foods, such as berries, mushrooms, turmeric, and garlic, has been shown to play a role in brain resilience by regulating neuroinflammatory signaling pathways [126]. The synergistic action of the different components present in these foods attenuates neuroinflammation and may help prevent neurodegeneration [126].

#### 3.1.1. Potential Anti-Neuroinflammatory Effects of Genistein

Recent studies showed that a dose of 10 mg/kg of GEN administered for 2 weeks decreased TNF-α levels and improved cognitive dysfunction in experimental models [127]. Moreover, GEN (20, 40, or 80 mg/kg/day) reduced isoflurane-induced apoptosis and neuroinflammation by inhibiting the TLR4 pathway in the rat hippocampus and in BV2 cells [128]. Furthermore, GEN (10 mg/kg, i.p.) attenuated neuroinflammation by downregulating NF-κB signaling pathways in brain tissues of neonatal mice subjected to hypoxic–ischemic brain damage [129].

#### 3.1.2. Potential Anti-Neuroinflammatory Effects of Fisetin

Fisetin enhanced autophagy in pro-inflammatory microglial cells via activation of the adenosine 5′-monophosphate-activated protein kinase (AMPK) pathway and inhibition of the mTOR signaling pathway, thereby mitigating neuroinflammation and reducing apoptotic effects in neurons [130]. Furthermore, fisetin (20 mg/kg) inhibited neuroinflammation by decreasing the expression of interleukin 1 receptor, type I (IL-1R1), pNF-κB, TNF-α, and iNOS in microglia, blocked NLRP3 inflammasome activation by promoting mitophagy through suppression of the secretion of IL-1β into CNS in vitro, and contributed to the amelioration of cognitive impairment in vivo [131]. In addition, fisetin suppressed activation of TLR4, myeloid differentiation factor 88 (MyD88), and NF-κB and subsequently inactivated pro-inflammatory cytokines including IL-6 and TNF-α. It can also reduce the accumulation of p-tau and Aβ, increase the expression of the Aβ remover neprilysin (NEP), and promote Pb-induced autophagy in the brains of mice via activation of the AMPK/Sirt1 pathway [132]. Pharmacological network studies revealed that fisetin binds to the CD44 protein, mitigating the neuroinflammatory response mediated by the I-kappa B kinase (IKK)/NF-κB signaling pathway and preventing neuronal damage in AD [133].

#### 3.1.3. Potential Anti-Neuroinflammatory Effects of Coffee and Chlorogenic Acids Metabolites

Interestingly, coffee and chlorogenic acid polyphenol metabolites such as 3′,4′-dihydroxycinnamic acid and 5-O-caffeoylquinic acid protected against early-life stress-induced cognitive deficits, potentially mediated by hippocampal neurogenesis associated with microglial changes [134]. Also, a secondary metabolite known as pentadecyl-2-oxazoline found in green and roasted coffee beans induced an enhancement of dopamine release in the medial prefrontal cortex and selectively normalized cortical GABA levels, restoring behavioral changes, including anxiety, depression, social interaction impairment, and aggressiveness [135].

#### 3.1.4. Potential Anti-Neuroinflammatory Effects of Anthocyanins

Red cabbage anthocyanin-rich extract and cyanidin-3-diglucoside-5-glucoside-rich extract, administered orally for 12 weeks, significantly reduced neuroinflammatory IL-1β and IL-6 cytokines in serum and brain by upregulating the MAPK signaling pathway and the abundance of butyrate-producing bacteria, thus preserving the functional profile of the microbial community [136]. Moreover, cyanidin- and delphinidin-rich supplementation at doses ranging from 2 to 40 mg mitigated the negative effects associated with high-fat diet consumption and obesity in the mouse hippocampus by decreasing neuroinflammatory mediators including TLR4, TNF-α, and Il-1β, improving glucocorticoid metabolism, and upregulating BDNF levels [137]. Oral administration of cyanidin-3-O-glucoside (30 mg/kg/day) anthocyanin inhibited neuronal apoptosis and tau protein phosphorylation and enhanced autophagy machinery and neuronal plasticity in the cerebral cortex and hippocampus of an APPswe/PS1ΔE9 transgenic mouse model by targeting the AMPK/Sirt1 pathway [138]. A recent study observed the anti-neuroinflammatory effects of anthocyanin-rich bioactive fraction combined with alginic acid nanocomplex (100, 200, and 400 μg/mL) by suppressing inflammatory cytokines such as TNF-α and IL-6 induced by lipopolysaccharide and Aβ aggregation in vitro. In addition, the nanocomplex also alleviated cognitive impairment in scopolamine-induced dementia in a murine model via stimulation of M2 microglial polarization in a concentration-dependent manner [139]. Overall, the data indicate that neuroinflammation accelerates neurodegeneration and cognitive decline; this pathological response can be prevented or reversed by neuronutrients, particularly polyphenols targeting anti-inflammatory pathways and brain resilience genes.

## 4. Metabolomics for Studying Nervous System Disorders: Personalized Neuronutritional Medicine

### 4.1. Tryptophan, Kynurenine, and Serotonin Metabolic Pathways

Tryptophan (Trp) is an essential aromatic aminoacidic, which can be converted into functional molecules such as serotonin and kynurenine (KYN), both important in regulating the neuronal activity of the nervous system [140]. In humans, Trp is exclusively obtained via dietary sources; therefore, its physiological levels can be assessed by controlling intake and the activity of several Trp metabolic pathways [141]. Increasing evidence has revealed that dysregulation of Trp metabolism plays a critical role in developing neuropathological processes such as episodic memory, cognitive decline, social behavior, aggression, and impulsivity [142,143]. Trp metabolism mainly requires three metabolic pathways named kynurenine, serotonin, and indole pathways (Figure 2) [144]. Trp is metabolized from the intestinal tract and subsequently transported across the BBB into the CNS via large neutral amino acid transporters LAT1 and LAT2 [145]. In normal conditions, the major Trp metabolism pathway is the kynurenine pathway (KP); through this pathway, >95% of Trp degrades in multiple neuroactive compounds [146]. First, Trp is converted into N-formyl kynurenine by indoleamine 2,3-dioxygenase (IDO1 or IDO2) and tryptophan-2,3-dioxygenase (TDO) and then to KYN. Specifically, TDO, IDO1, and IDO2 are the crucial rate-limiting enzymes in the KP producing downstream metabolites quinolinic acid (QUINA) and kynurenic acid (KYNA) (Figure 2) [147]. Notably, IDO1 transported from the blood across the BBB plays a pathogenic role in the CNS when activated by neuroinflammatory cytokines such as interferon γ (IFN-γ), IL-6, and TNF-α [148]. The latter disrupt the BBB and allow increased expression of IDO1 and other factors, which in turn contribute to the worsening neuroinflammation and subsequent brain dysfunction [148]. Recent studies observed that KYNA significantly suppressed LPS-induced macrophage pyroptosis by reversing the expression of NOD-like receptor protein 3 (NLRP3), Gasdermin-D, and Caspase1 and the expression of inflammatory factors via activating the Nrf2 pathway and brain resilience proteins HO-1 and GSH [149]. Similarly, other evidence reported that KYNA prevented changes in Nrf2 levels, oxidative imbalance, and mitochondrial dysfunction caused by QUINA toxicity in striatal slices, Table 2 [150]. KP Metabolites, particularly 3-HK and XANA, are known to be neurotoxic, whereas KYNA and PA are usually characterized as neuroprotective (Figure 2). The 5-hydroxytryptamine (5-HT) or serotonin pathway is involved in various physiological processes and plays important roles throughout the body. Trp can be converted to 5-hydroxytryptophan (5-HTP) and 5-HT in central neurons and enterochromaffin cells (Figure 2) [151]. 5-HT is an important neurotransmitter involved in regulating adaptive responses and environmental changes, such as sleep, cognition, and feeding behavior (Figure 2) [151]. Of note, Trp and its neuroactive metabolites play a central role in the crosstalk along the gut–brain axis [152]. Indeed, gut microbes can directly convert Trp into various molecules, such as indole and its derivatives (indole-3-aldehyde (IAld), indole-3-acetic-acid (IAA), and indole-3-propionic acid) [153]. Indole and its derivatives maintain intestinal homeostasis by regulating the expression of proinflammatory and anti-inflammatory cytokines. Interestingly, recent evidence by Wojciech et al. reported that alterations in CD^4+^T intestinal cells depended on the Blastocystis infection-derived indole-3-acetaldehyde (I3AA) metabolite. The latter activated pro-inflammatory effectors, and antagonized the intracellular microbiota–aryl hydrocarbon receptor (AhR), exacerbating ulcerative colitis in vitro and in vivo in a dose-dependent manner [154]. Overall, we postulate that a personalized neuronutritional approach targeting Trp metabolism, which in turn leads to activation of the Nrf2 pathway, promotes brain resilience and could provide a novel and potentially advantageous therapy to prevent and treat nervous system disorders.

#### 4.1.1. Tryptophan Metabolites in AD

Dysregulation of Trp metabolites was suggested as a major contributor to AD pathogenesis [155]. With respect to the neuropathophysiology of AD, KYNA is considered neuroprotective because it is an antagonist for ionotropic glutamate receptors as well as the α7 nicotinic acetylcholine (α7nACh) receptor [155]. On the contrary, QUINA is considered neurotoxic because it is an agonist of the N-methyl-D-aspartate (NMDA) receptor with the potential to induce excitotoxicity [155]. Recent evidence has shown a neuropathological profile of AD through crosstalk between inflammatory and KYN pathways, generating a range of neurotoxic metabolites with degenerative effects, in particular QUINA and IDO, both correlated to altered inflammatory profiles and cognitive decline in AD mice infected by encephalitis virus [156]. Specifically, peripheral IL-6-associated microglial QUINA elevations in the basolateral amygdala contributed to cognitive decline in the model of postoperative delirium [157]. Increased levels of IDO and 3-HK have been reported in the brains of patients with AD when compared to controls [158]. Of note, the association of IDO-1 with senile plaques was confirmed and IDO-1 was shown to be specifically localized in conjunction with neurofibrillary tangles. Therefore, amyloid-β, through its activation of microglia and astrocytes, induced the upregulation of the neurotoxic intermediates of the KP that produced further oxidative stress excitotoxicity, and tau phosphorylation. Consistent with the induction of IDO expression by amyloid-β, higher QUINA levels were also observed in microglial cells in proximity to amyloid plaques with neurofibrillary tangles [159]. This confirms that QUINA induced tau phosphorylation in a culture of primary human neurons [159]. In addition, significantly higher TDO and IDO1 levels in both human and mouse hippocampus lead to the excessive formation of KYN metabolites such as QUINA closely involved in the neurodegenerative processes in AD [160]. Interestingly, in astrocytes and neurons from AD subjects, IDO1 inhibition enhanced astrocytic production of lactate and uptake by neurons. Moreover, Aβ- and tau-dependent increases in IDO1 and KYN levels could be blocked by the small-molecule IDO1 inhibitor PF06840003 or through genetic deletion of IDO1 [161]. Thus, inhibition of IDO1 enhanced hippocampal glucose metabolism and memory function in a mouse model of AD by restoring astrocyte metabolism [161]. Moreover, epigenetic alterations in Trp- and NAD-pathway-associated genes including SIRT1, PARP9, PARP14, QPRT, PARP10, NAPRT, NADSYN1, and NADK showed a positive correlation in both plaque and tangle, while PNP, NT5C2, PARP1, and PARP4 were positively correlated in plaque alone. Importantly, DNA methylation in cg11251498 (IDO2), an immune regulatory gene, has been identified as a key candidate target in AD [162]. Overall, genetic or pharmacological inhibition of IDO and QUINA expression may be effective in suppressing AD progression.

#### 4.1.2. Tryptophan Metabolites in PD

In PD patients, Trp concentrations were lower, and the KYN: Trp ratio, KYN, and anthranilic and kynurenic acids were higher than in controls [155]. Notably, 3-HK was elevated in the brains of patients with PD, whereas KYN and KYNA were decreased, indicating the contribution of 3-HK neurotoxicity to the pathogenesis and its potential as a predictive marker of PD [163]. More recently, a study performed on 100 participants with PD and 90 healthy controls indicated that higher levels of neurotoxic QUINA/KYNA ratio in both plasma and CSF were strongly correlated with peripheral and brain inflammation, which ultimately exacerbated motor symptom severity and PD progression [164]. Interestingly, serum levels of KYN and beta-alanine were higher in PD patients than in the control group and this was correlated to the reduction in Wnt pathway genes in leukocytes [165]. Pharmacological inhibition of IDO, KMO, and kynureninase (KYNU) in microglia may be targeted to protect against reactive microglial-associated neuronal atrophy [166]. A very recent study observed significantly higher levels of neurotoxic QUINA and 3-HK in plasma and lower concentrations of neuroprotective KYNA along with higher neurotoxic QUINA/KYNA ratios in the CSF of PD patients. Specifically, the authors found a pattern of KP dysregulation correlated with increased severity of PD symptoms (Figure 2), particularly evident in women [167]. A recent analysis showed that PD patients who presented with milder motor symptoms had higher levels of xanthurenic acid and a higher XA/3-HK (3-hydroxykynurenine) ratio. Similarly, reduced levels of neurotransmitters such as serotonin (5-HT) and GABA were associated with more pronounced non-motor symptoms These findings indicate that alterations in neurotransmitter levels and their conversion metabolism could serve as potential biomarkers to aid in the diagnosis of PD [168]. Moreover, Chung et al., using the LC-MS/MS method, have developed 21 tryptophan metabolites within the indole, KYN, and serotonin metabolic pathways in human urine samples. The authors found aberrant urinary levels of indole-3-acetic acid in PD patients (Figure 2) [169]. Finally, a clinical study investigated the impact on kynurenine 3-monooxygenase (KMO) gene mutations in late-onset PD patients. Genomic analysis revealed a high prevalence of missense mutations in the late-onset PD groups leading to a reduction in 3-HK levels and limited progression of PD pathogenesis [170]. Equally important, Fargher et al., using CRISPR/Cas9, have generated a zebrafish model of α-amino-β-carboxymuconate-ε-semialdehyde decarboxylase gene (ACMSD) deficiency. This mutation in the ACMSD gene upregulated the neuroprotective KYNA, an NMDA antagonist, which in turn protected against QUINA-induced toxicity [171]. Emerging evidence has reported that AHR, a ligand-dependent transcription factor, induced parkin expression. KYN, being an endogenous AHR ligand, promoted neuroprotection in PD [172]. Notably, KYN treatment inhibited neuronal apoptosis and excitotoxicity mediators activated by rotenone exposure, in particular alpha amine-3-hydroxy-5-methyl-4-isoxazol propionic acid and N-methyl-D-aspartate receptors in dopaminergic neurons [172]. Finally, systematic metabolomic studies identified elevated levels of acetyl phenylalanine and tyrosine as well as reduced levels of KYN in the urine of early- and mid-stage PD patients related to disease progression (Figure 2) [173].

#### 4.1.3. Tryptophan Metabolites in Autism

Dysregulation in the Trp metabolic pathways and abundance indoles, KYN and particularly serotonin, has been associated with behavioral deficits and ASD [174]. Recent evidence revealed that gut microbial alterations specifically associated to tryptophan fecal-related metabolites (i.e., indoles, serotonin) are implicated in atypical activity in interoceptive brain regions and ASD pathophysiology in youth (Figure 2) [175]. Of note, fecal tryptophan metabolites can cross the intestinal barrier and the BBB via vagal signaling, causing behavioral and cognitive alterations. Likewise, the same authors observed lower levels of kynurenate (KA) in an ASD group compared to neurotypical children [175]. In the gut microbiota, more than 90% of dietary tryptophan is primarily converted into KYN and to a lesser extent into serotonin and other metabolites. It is important to note that an imbalance in the regulation of the KP, in particular reduced plasma KA levels and an increased KYN/KA ratio, can lead to neurotoxic damage occurring with low blood serotonin levels and subsequently cognitive and behavioral alterations [176]. Equally important, prenatal exposure to antibiotics has been shown to disrupt the maternal microbiome and alter the levels of microbial metabolites which can then be transferred to the fetus and potentially affect brain development [177,178]. The effects of maternal stress on behavior and cerebral Trp metabolism in offspring vary by sex. This suggests that the emotional dysfunction observed in mice exposed to prenatal stress (during in utero development) may be caused by alterations in both the 5-HT and KYN pathways.

Specifically, females exposed to prenatal stress exhibited depression-like behavior, mainly when in proestrus/diestrus, with reduced 5-HT levels in the hippocampus and an increased turnover rate in the hippocampus and brainstem. Males exposed to prenatal stress, however, exhibited anxiety-like behavior and higher levels of QUINA in the hippocampus and brainstem compared to male offspring controls [179].

In addition, it has been observed that an alteration of the serotonin pathway can increase the risk of ASD, while an increase in choline metabolites can reduce it (Figure 2) [180]. Both chemogenetic activation of dopaminergic neurons and methylphenidate treatment ameliorated ASD-like changes in a mouse model of IDO2 KO mice [181]. A recent clinical study demonstrated that serum 3-hydroxykynurenine and KYNA concentrations were significantly higher in the ASD group than in the control group, whereas serum 3-hydroxyanthranilic acid concentrations were significantly lower in both toddlers and preschool children with ASD [182]. Another recent clinical trial reported that KP was altered at various levels in ASD. Accordingly, in ~58% of the cases, individuals with ASD showed low-grade chronic inflammation that is primarily driven by chronic AhR activation. This persistent inflammatory state is often associated with a high prevalence of gastrointestinal disorders and IDO activation [183]. This mechanism of neurotoxic action also suggests that the KYN/TRP ratio could be considered a novel biomarker for evaluating and quantifying low-grade inflammation in ASD pathophysiology [183]. Finally, Yildirim and coworkers showed that serum levels of KYNA/QUINA, KYN, and IL-6 were significantly higher in the ASD children than in the controls [184].

### 4.2. Pharmacological Inhibitors and Neuronutrients Modulate Tryptophan Metabolites

#### 4.2.1. Preclinical Studies

Recent preclinical evidence has shown that neuronutrients modulate Trp metabolites. Indeed, metabolomic analysis reported that supplementation with a dose of 30 mg of quinic acid improved high-fat diet (HFD)-induced neuroinflammation and oxidative stress, by downregulating the DR3/IKK/NF-κB signaling pathway and upregulating IAA and KYNA metabolites [185]. Due to the low bioavailability of polyphenols, most dietary polyphenols exert their function through microbial metabolites fermented by gut microorganisms. Accordingly, changes in microbial metabolites after quinic acid treatment indicated a significant upregulation of CAT, GPx, SOD, and IL-10, and a downregulation of MDA, IL-1β, IL-6, Aβ42, and Tau in vivo [185]. Another polyphenol known as ferulic acid suppressed LPS-induced IDO expression, mediated by the inhibition of the NF-κB and p38 MAPK pathways in microglial cells [186]. Also, apigenin at a low dose of 20 μM inhibited pro-inflammatory IL-6 and nitric oxide production, the activation of ERK, JNK and MAPK, and the degradation of IκBα in LPS-stimulated microglial cells dose-dependently [187]. A recent study demonstrated that sertraline, tiagabine, and bicifadine exhibited autophagy activity and neuroprotection by inhibiting the mammalian target of rapamycin (mTOR) and restoring the quantity of cellular neurotransmitters such as betaine, 5-hydroxyindoleacetic acid, and KYN in 6-OHDA-induced neurotoxicity in PC12 cells [188]. Of interest, a new pharmacological compound named CZ-17 showed moderate inhibitory activity to IDO1 and TDO, respectively. CZ-17 potentially crossed the BBB via passive diffusion, inhibited the KP at the cellular level, and remarkably reduced the KYN/TRP ratio. CZ-17 displayed a direct neuroprotective effect in a corticosterone-induced PC12 neural cell injury model. In in vivo experiments, CZ-17 significantly increased dopamine and serotonin levels, improved MPTP-induced motor disability, and rescued LPS-induced depressive behavior in a zebrafish model (Figure 2) [189].

Furthermore, the probiotic *Bifidobacterium longum* (*B. longum*) CCFM077 was observed to improve autism-related behaviors, such as stereotyped and repetitive behaviors, and to enhance learning and memory skills. This effect is due to its ability to regulate kynurenine (KYN) metabolism in the gut, blood, and brain. Specifically, B. longum CCFM1077 increased brain levels of QUINA, glutamic acid (Glu), and the Glu/γ-aminobutyric acid (GABA) ratio, and improved the activity of microglia in the cerebellum.

Data analysis showed a strong correlation between elevated QUINA levels in the brain and the presence of autistic behaviors, suggesting a link between this metabolite and the levels of excitatory and inhibitory neurotransmitters such as GABA and Glu [190].

#### 4.2.2. Clinical Studies

In humans, supplementation with vitamin D_3_ improved physical performance by enhancing the neuroprotective metabolite KYNA and decreasing the neurotoxic metabolite 3-HK in PD patients after 12 weeks [191]. Recent research found an increased neuroexcitatory QUINA/KA ratio in both plasma and CSF of PD participants associated with peripheral and cerebral inflammation and vitamin B_6_ deficiency. Furthermore, increased QUINA tracked with CSF tau, CSF-soluble TREM2 (sTREM2), was correlated to the severity of both motor and non-motor PD clinical symptoms [192]. A randomized, controlled, crossover trial in older adults revealed that dietary polyphenols significantly increased the serum concentration of indole 3-propionic acid, a metabolite generated by the microbial degradation of dietary tryptophan, associated with changes in C-reactive protein and gut microbiome bacteria, particularly increasing the abundance of Clostridiales and decreasing the abundance of Enterobacteriales [193]. The administration of Bifidobacterium BB-12 significantly increased the levels of glutamine, glutamic acid, and KYN and reduced the levels of pro-inflammatory markers including TLR4, NF-κB, IL-1β, and TNF-α in the serum, thus alleviating systemic inflammatory processes in the gut of premature infants [194]. Finally, a randomized controlled trial observed that healthy volunteers that received active doses of a new pharmacological inhibitor GSK3335065 (1.3 mg) or placebo showed partial arrest of the KMO metabolite [195].

### 4.3. SCFAs Metabolism Along the Gut–Brain Axis: Focus on Nutrients

The gut microbiota represents a large microbial community, comprising bacteria including Bacteroidetes, Proteobacteria, Actinobacteria, Firmicutes, Verrucomicrobia, and Fusobacteria, with Firmicutes and Bacteroidetes being the most abundant species [196]. Intestinal bacteria perform various functions, including digestion of foods, degrading complex carbohydrates and insoluble fibers introduced with the diet (mainly Bacteroidetes and Firmicutes), and the synthesis of vitamins (e.g., thiamine, biotin, riboflavin, pantothenic acid, folate, vitamin K), amino acids, antimicrobial substances, and particularly short-chain fatty acids (SCFAs) [197]. The latter are the main metabolites produced by colon bacterial fermentation of dietary fiber and are speculated to play a pivotal role in gut–brain crosstalk [198]. The most abundant SCFAs in the colon are acetic acid (C2), propionic acid (C3), and butyric acid (C4) [198]. These metabolites are absorbed in the colon and transported to the liver via the H+-dependent or sodium-dependent monocarboxylate transporters (MCTs and SMCTs), and only a small fraction reaches the systemic circulation and tissues [199]. SCFAs exhibit beneficial effects in maintaining intestinal barrier integrity, supporting gut mucosa and protecting against inflammation from the gut to the neuro-immune system [200,201]. In the brain, SCFAs cross the BBB to directly activate neurons by binding to G protein–coupled receptors mediated by the stimulation of the Nrf2 pathway [3,202]. Importantly, SCFAs reinforce the integrity of the BBB, which is closely associated with the controlled passage of certain molecules and nutrients from the systemic circulation to the brain, promoting neuronal development and preserving CNS homeostasis. Supporting the notion that SCFAs regulate BBB function and integrity, a study performed by Braniste and colleagues in germ-free (GF) mice has shown that reduced expression of tight junction proteins (claudin and occludin) leads to enhanced BBB permeability from intrauterine life to adulthood [203]. In essence, recolonizing GF mice with a complex microbiota or with SCFA-producing bacterial strains, particularly sodium butyrate, recovered BBB integrity [203]. Similarly, a physiological dose of propionate treatment (1 μM) attenuated brain endothelial permeability after LPS exposure via Nrf2 signaling in vitro [204]. Moreover, the neuroprotective effect of Clostridium butyricum showed significant improvements in neurological dysfunction, brain edema, neurodegeneration, and BBB impairment by increasing the expression of occludin and zonula occluden-1, p-AKT, Bcl-2, and glucagon-like peptide-1 (GLP-1) secretion, as well as decreasing the expression of Bax in TBI mice via the gut–brain axis [205]. SCFAs help keep the intestine healthy, protecting it from inflammation by regulating the production of IL-6, IL-8, IL-12, IL-17, IL-1β, and TNF-α by colon epithelial cells. Furthermore, SCFAs have a range of immunoregulatory properties [197]. The production of short-chain fatty acids is entrusted to the bacteria resident in the intestinal mucus, mainly represented by Akkermansia muciniphila, Bifidobacterium bifidium, Bacteroides fragilis, and Ruminoccous gnavus. These bacterial species are capable of degrading mucus through a series of enzymes (glycosidase, sialidase, galactosidase, neuramidase, sulfatase, cysteine protease) which use glycans as an energy source, producing SCFAs through the fermentation process [206]. Furthermore, the monosaccharides produced by cleavage of mucus O-glycans are used by other resident bacteria such as Enterobacteriaceae, Lachnospiraceae, and Clostridium cluster XIV. The production of butyrate with a protective action on the intestine by intestinal clostridia is widely documented [207]. The production of SCFAs induces important antioxidant, anti-inflammatory, and barrier-protective responses. Ultimately, epigallocatechin-3-gallate determines the reduction in intestinal permeability, improving tight junctions, thus reducing inflammation and intestinal damage [208]. The results provided by in vitro and in vivo studies have demonstrated an increase in beneficial bacteria, such as Lactobacillus spp. and Bifidobacterium spp. following the intake of foods rich in anthocyanins. Furthermore, they counteract pathogenic bacteria by increasing the levels of SCFA with antimicrobial action [209]. In particular, it was observed that anthocyanins extracted from Opuntia ficus-indica significantly increased the content of SCFAs in the intestines of mice. Among these, the greatest increase was found in acetic acid, propionic acid, and butyric acid. This suggests that anthocyanins are able to modify the diversity and composition of the intestinal bacterial flora, thus promoting the production of SCFAs [209].

#### 4.3.1. Neuronutrients Regulate SCFA Metabolites in Nervous System Disorders: SCFA Metabolites in AD

##### Preclinical Studies

SCFAs, as neuroactive messengers characterized by the ability to penetrate the BBB, have recently received unprecedented attention in the pathogenesis of AD. The importance of the adherence to a diet rich in fiber and SCFAs in attenuating AD biomarkers, slowing cognitive decline, and consequently the risk of AD incidence has been widely highlighted [210]. Indeed, depletion of fiber/SCFAs accelerated memory deficits, whereas diets supplemented with high acetate and butyrate delayed cognitive decline, and impaired adult hippocampal neurogenesis in AD mice. Moreover, maternal dietary fiber intake significantly altered offspring’s cognitive functions in 5xFAD mice and microglial transcriptome, suggesting that SCFAs may exert their effect during pregnancy and lactation [211]. Exogenous ketone supplementation containing β-hydroxybutyrate positively modulated β-site amyloid precursor protein cleaving enzyme 1 (BACE1) activity in C57BL/6J mice [212]. Metabolomic analysis of murine feces suggested that gut microbiota butyrate-producing bacteria are able to prevent cognitive decline and the development of AD by mitigating the deleterious impact of the APOE genotype [213]. Recent research demonstrated that Akkermansia muciniphila and its metabolite propionate reduced mitochondrial fission protein (DRP1) via G protein–coupled receptor 41 (GPR41), increased PINK1/PARKIN-mediated mitophagy via G protein–coupled receptor 43 (GPR43), and protected BBB integrity, thereby ameliorating cognitive dysfunction in hyppocampal neurons and rodent models of AD [214]. Decreased concentrations of SCFA and other bacterial metabolites appear to play a major role in the onset of neurocognitive symptoms in AD and PD. Increased abundance of proinflammatory taxa could be closely related to the more severe clinical symptoms in dementia. Moreover, geographical differences in the composition of the gut microbiota have been reported in AD. Some potential beneficial effects of probiotics in AD and PD have been reported [215]. Fecal metabolomic and microbiomic analyses revealed that Schisandra chinensis polysaccharide significantly increased the content of SCFAs, reversing gut microbiota disorders in AD rats [216].

A recent study by Xiao-Hang et al. [217] examined the neuroprotective effects of two probiotic mixtures on aged mice: Probiotic-2 (P2), composed of Bifidobacterium lactis and Lactobacillus rhamnosus; and Probiotic-3 (P3), composed of Bifidobacterium lactis, Lactobacillus acidophilus, and Lactobacillus rhamnosus. Administration of these probiotics (1 × 10^9^ colony-forming units) once daily for 8 weeks led to an increase in SCFAs, particularly valeric, isovaleric, and hexanoic acids. This increase was correlated with several beneficial effects, including improved cognitive function, reduced neural damage, less amyloid β- and Tau-related pathology, and decreased neuroinflammation. These findings suggest that probiotics act by modulating the activation of the AKT/GSK-3β signaling pathway in the brain [217]. Moreover, a Traditional Chinese medicine named Coptis chinensis Franch regulated potential metabolites, particularly increasing the content of ILA (indole-3-lactic acid) and SCFAs in AD mice via the gut–brain axis [218]. Furthermore, anthocyanins have shown brain resilience effects by potentially mitigating cognitive impairment and aberrant amyloidogenesis through increased SCFAs content for gut and brain homeostasis [219]. In addition, a triterpenoid saponin known as xanthoceraside was demonstrated to improve AD rats’ learning and memory deficits by modulating the community of gut microbiota and SCFAs production in vitro and in vivo [220]. Also, palmatine treatment had the potential to enhance the content of SCFAs and reverse gut microbiota alterations through the activation of AMPK autophagy signaling and the inhibition of the mTOR pathway in AD murine models [221]. Importantly, encapsulated lactiplantibacillus plantarum effectively reduced Aβ deposition and tau protein phosphorylation and improved the neuroinflammatory response by enhancing colon SCFA levels in APP/PS1 transgenic mice [222].

##### Clinical Studies

Unfortunately, to date, only a few studies have investigated the efficacy of SCFA metabolites in inhibiting the onset and progression of AD in humans. A recent randomized controlled trial conducted on 63 healthy young adults who consumed 25 g/day of peanut polyphenols observed a significant increase in fecal SCFAs, plasma, and fecal very long-chain saturated fatty acid (VLCSFA) levels associated with ameliorated memory functions and stress response mechanisms [223].

### 4.4. SCFA Metabolites in PD

#### 4.4.1. Preclinical Studies

Recent evidence suggests that neuronutrients activating SCFA metabolites show promising therapeutic potential to inhibit PD pathogenesis. Indeed, disturbances in the metabolism of dietary peripheral branched-chain amino acids (BCAAs) can contribute to neuroinflammation and the development of PD. Specifically, BCAAs supplementation improved both motor and non-motor symptoms and elevated SCFA production, in particular propionic acid concentrations, while decreasing isovaleric acid concentrations in a mouse model of 1-methyl-4-phenyl-1,2,3,6-tetrahydropyridine (MPTP)-induced PD [224]. This postulates that propionic acid may be a potential therapeutic biomarker for PD [224]. Conversely, other recent studies showed that there exists a critical link between gut-derived metabolic changes and neuroinflammatory processes in PD via targeting the SCFA/GPR43-NLRP3 pathway [225]. Notably, SCFAs can exacerbate motor and gastrointestinal dysfunctions in PD models, intensifying α-syn pathology and neuroinflammation in vitro and in vivo [225]. Interestingly, a study found that sodium butyrate promoted α-synuclein degradation mediated by Atg5-induced autophagy via the PI3K/Akt/mTOR pathway in murine neuroendocrine STC-1 cells, Table 2 [226]. Similarly, the concentrations of acetate, butyrate, and especially propionate were significantly downregulated in the fecal samples of PD patients. Propionate administration improved motor behavior and the intestinal epithelial barrier by increasing the protein expression of zonula occludens-1 and occludin in MPTP-induced PD mice via the activation of the protein serine-threonine kinases (AKT) signaling pathway [227].

#### 4.4.2. Clinical Studies

A recent analysis of fecal metabolites in patients with PD displayed altered gut microbiota composition and metabolites compared to healthy subjects. Specifically, the PD fecal samples showed higher levels of cadaverine, ethanolamine, hydroxypropionic acid, isoleucine and leucine, phenylalanine, and thymine metabolites. In contrast, the levels of butyric acid, propionic acid, acetic acid, linoleic acid, oleic acid, nicotinic acid, glutamic acid, pantothenic acid, pyroglutamic acid, succinic acid, and sebacic acid were significantly decreased, which was associated with reduced abundance of Lachnospiraceae bacterial species [228]. An open-label, non-randomized study demonstrated that prebiotic fiber interventions decreased severity of motor and non-motor PD symptoms and ameliorated gut function in patients. In particular, the neuroprotective effects of prebiotic fibers have been associated with adequate production of SCFA metabolites in plasma, a reduced abundance of pro-inflammatory bacteria such as Escherichia coli, and an increased abundance of beneficial bacteria such as Prevotella, the families Lachnospiraceae and Ruminococaceae (regulated by resistant starch), Ruminoccocus, Dorea, Bacteroides (regulated by rice bran), Blautia, Anaerostipes, and Bifidobacterium (regulated by inulin), and the genus Parabacteroides (regulated by resistant maltodextrin) [229]. A prospective, controlled pilot study observed that short-term prebiotic dietary interventions, including the intake of the prebiotic Lactulose, promoted changes in gut metabolites. The dietary fibers significantly increased fecal and urine SCFA metabolites and the relative abundance of *Bifidobacteria* spp., reducing PD-related gastrointestinal symptoms and upregulating S-adenosyl methionine, GSH, and inositol in PD patients [230]. Moreover, an interesting study revealed lower levels of SCFAs and higher levels of calprotectin in the feces of PD patients. Specifically, lower SCFA levels and higher butyric acid levels were correlated with the age of onset for motor and non-motor PD symptoms. Conversely, a significant reduction in plasma levels of the chemokine CXCL8 was found in female PD patients. However, an inverse correlation between fecal CXCL8 and IL-1β and non-motor symptom severity, particularly gastrointestinal inflammation related to PD progression, has been also observed [231].

### 4.5. SCFA Metabolites in Autism

#### 4.5.1. Preclinical Studies

Recent evidence has shown that chronic oral treatment with acetyl-L-carnitine improved social interaction deficiency and repetitive behavior by enhancing gut microbiota SCFA (i.e., acetic acid and butyric acid) levels and reducing TNF-α and IL-1β in valproate-induced autistic rats [232]. Likewise, prenatal exposure to valproic acid significantly altered SCFAs, i.e., reduced acetic acid, propionic acid, butyric acid, valeric acid, hexanoic acid, isobutyric acid, and isovaleric acid metabolites in feces, and some neurotransmitters, i.e., kynurenine, tryptophan, threonine, 5-HIAA, and BAC in the prefrontal cortex of rats via the gut–brain axis. Therefore, SCFAs supplementation (butyric acid) could be a potential therapeutic target for regulating central neurotransmitter metabolism related to changes in beneficial gut bacteria [233]. Finally, oral treatment with acetate reversed Shank3 genotype and microbiome depletion, ameliorating social deficits in knockout mice [234].

#### 4.5.2. Clinical Studies

Nutritional therapeutic strategies through butyrate-producing bacteria modulating gut microbiota to relieve ASD symptoms have been observed in humans [235]. Consistent with this, a clinical trial performed on 26 children with ASD supplemented with probiotics + fructo-oligosaccharide or placebo has shown a significant reduction in the severity of autism and gastrointestinal symptoms associated with increased levels of SCFA metabolites such as acetic acid, propionic acid, and butyric acid levels, increased levels of homovanillic acid, and decreased levels of serotonin correlated with abundance in beneficial bacteria (Bifidobacteriales and *B. longum*) and suppression of the pathogenic bacteria Clostridium [236]. It is important to consider that adequate intake of functional nutrients influences SCFA concentrations, specifically acetate, propionate, butyrate, isovalerate, and valerate in ASD children after a 6-month period (Figure 2) [237].

### 4.6. Sulfur-Containing Nutrient Metabolites

Sulfur-containing metabolites are known as an essential class of plant secondary metabolites, with potent neuroprotective properties. Recently, these metabolites have shown to provide suitable advantages in preventing and managing nervous system disorders [238].

#### 4.6.1. Dihydroasparagusic Acid

Dihydroasparagusic acid (DHAA) is the reduced form of asparagusic acid, a sulfur-containing flavor component produced by Asparagus plants, and it effectively inhibits neuroinflammatory and oxidative processes that are primary factors for the etiopathogenesis of AD and PD.

Moreover, it significantly reduced the production of pro-inflammatory and neurotoxic mediators induced by LPS. Specifically, it decreased the expression of nitric oxide, tumor necrosis factor-α, and prostaglandin E2, Table 2. DHAA also decreased the expression of proteins such as inducible nitric oxide synthase and cyclooxygenase-2, and reduced lipoxygenase activity in microglia [239].

#### 4.6.2. S-Allyl Cysteine

S-Allyl cysteine, a sulfur-containing secondary metabolite with the chemical formula of C_6_H_11_NO_2_S, is present in significant amounts in fresh garlic. Compelling evidence has reported that S-Allyl cysteine (30 mg/kg) prevented free radical-associated deterioration of cognitive function and neurobehavioral impairments, inhibiting oxidative stress, GPx, and GSH in a mouse model of AD [240]. Furthermore, s-allyl cysteine showed neuroprotective and anti-amyloidogenic effects by activating the Nrf2 brain resilience response in vitro and in vivo [241,242,243]. Similarly, S-allyl-l-cysteine protected hippocampal and cerebellar granule neurons isolated from embryos of Wistar rats against the neurotoxicity induced by Aβ protein [244]. Additionally, sulfur-containing amino acid metabolites including s-allyl cysteine, s-ethyl cysteine, and s-propyl cysteine have been shown to decrease Aβ protein production and restore the brain enzymes GPX, SOD, and catalase [245]. Moreover, s-allyl cysteine treatment ameliorated cognitive deficits through modulation of the Nrf2/NF-κB/TLR4/HO-1 pathway, acetylcholinesterase, and inhibition of oxidative stress and neuroinflammation in streptozotocin-diabetic rats [246]. Finally, recent research reported that s-allyl-cysteine alone or in combination with temozolomide effectively reduced glioblastoma cells proliferation via the inhibition of the Nrf2 pathway and GSH levels in a dose-dependent manner [247].

#### 4.6.3. 6-(Methylsulfinyl)hexyl Isothiocyanate

6-(Methylsulfinyl) hexyl isothiocyanate (6-MSITC), a key bioactive compound found in wasabi, is another sulfur-containing metabolite that has demonstrated brain resilience potential in vitro and in vivo [248,249]. In experimental models, 6-MSITC preserved functional nigral dopaminergic neurons, which contributed to the reduction in motor dysfunction induced by 6-OHDA [248]. More recently, the same authors reported that a low dose of 5 mg/kg of 6-MSITC exerted significant neuroprotective effects in animals against the Aβ-induced ROS, cognitive impairment, and neuroinflammation by enhancing Nrf2/GSH activity [250]. Moreover, 6-MITC present in Wasabia japonica have anti-inflammatory activity for the treatment of IBD by inhibiting the GSK-3β/NF-κB pathway, Table 2 [251]. In a study performed by Uruno et al., administration of 6-MSITC improved the impaired cognition of AD mice via induction of Nrf2, which significantly suppressed oxidative stress and neuroinflammation [252]. A recent clinical study investigated the neuroprotective effects of 6-MSITC or placebo on seventy-two older adults. Specifically, 6-MSITC capsules (0.8 mg) containing 100 mg wasabi extract powder showed potential therapeutic activity to enhance working memory and episodic memory in older adults after 12 weeks [253].

#### 4.6.4. Hydrogen Sulfide

The neuroprotective properties of hydrogen sulfide (H_2_S) arise from its role as a gaseous signaling molecule that regulates neuron death or survival [254]. However, H_2_S is dysregulated during aging. Indeed, a recent study suggested that elevated plasma sulfides were associated with cognitive dysfunction and measures of microvascular disease in AD and related dementias [255]. The biosynthetic enzyme, cystathionine γ-lyase (CSE), is depleted in 3xTg-AD mice and in human AD brains [256]. H_2_S prevented hyperphosphorylation of Tau by sulfhydrating its kinase, glycogen synthase kinase 3β (GSK3β), while administering the H_2_S donor sodium GYY4137 (NaGYY) to 3xTg-AD mice improved motor and cognitive deficits in AD [256]. Excitingly, H2S increased neurogenesis by regulating the Akt/glycogen synthase kinase-3β/β-catenin signaling pathway in neural stem cells and in a mouse model of PD [257]. Moreover, NaHS (a donor of H_2_S) attenuated the cognitive impairment and promoted microglia polarization from M1 towards M2 in the hippocampus of PD rats [258]. Furthermore, H_2_S exposure reduced epigenetic histone deacetylases (HDAC) activity, suggesting its neuroprotective role in PD [259]. Of note, polyphenolic compounds including anthocyanins, resveratrol, and rosmarinic acid can oxidize H2S to polysulfides and thiosulfate, which contributes to their cytoprotective effects [260].

### 4.7. Tyrosine Metabolism and Neuronutrition: Tyrosine Metabolites in AD

#### 4.7.1. Preclinical Studies

Tyrosine metabolism regulates abnormal energy metabolism, reduces inflammation, and regulates gut flora and neurotransmitters in the treatment of neurodegeneration [261]. Consistent with this, the authors, using mass spectrometry-based urinary metabolomics, showed that the Radix ginseng-Schisandra chinensis herb significantly improved brain pathologic symptoms (Figure 3). Notably, they identified phenylalanine and tyrosine metabolites as potential endogenous biomarkers inducing neuroprotective effects [261]. A recent study using gut microbiota correlation analysis and metabolomics revealed that sika deer antler protein improved AD by modulating tyrosine metabolism and upregulating the PI3K/AKT/Nrf2 signaling pathway (Figure 3) [262]. A multi-omics and network analysis revealed the anti-AD effects of Kai-Xin-San in ameliorating cognitive deficits and pathological morphology of the hippocampus in AD rats by reducing oxidative stress, neurotoxicity, and neuroinflammation. A total of nine metabolites were identified, related to pyrimidine metabolism, riboflavin metabolism, tyrosine metabolism, tryptophan metabolism, and glycerophospholipid metabolism [263]. In vivo and in vitro metabolomic studies observed that p-coumaric acid regulated metabolic pathways in Aβ25–35-injected mice mainly involving arachidonic acid metabolism, tyrosine metabolism, unsaturated fatty acid biosynthesis, glycolysis/glycogenesis and glycerophospholipid metabolism, by blocking nuclear translocation of NF-κB to improve cognitive deficits targeting the activation of PI3K/AKT/Glut1 and inhibition of MAPK/NF-κB signaling, Table 2 [264]. Metabolomic results indicated that polygala tenuifolia polysaccharide regulated metabolites involved in sphingolipid metabolism, glycerophospholipid metabolism, tyrosine metabolism, and arachidonic acid metabolism by increasing the relative abundance of bacteria such as Alistipes and Lachnospiraceae [265]. Consumption of defatted walnut powder has been shown to improve spatial learning and memory, enhance cholinergic function, and reduce histopathological damage in the cortex and hippocampus of AD model mice. Metabolomic analysis identified specific metabolites and metabolic pathways modulated by defatted walnut powder, particularly those related to cellular energetics and antioxidant capacity. Interestingly, increased levels of arginine, histidine, proline, serine, and tyrosine were observed, along with a concomitant decrease in glutamate, thus mitigating the neuropathology and cognitive dysfunction induced by scopolamine-induced AD [266]. Urolithin A, a metabolite derived from ellagic acid and ellagitannin through the intestinal flora, has shown neuroprotective effects. The main target of urolithin A is the dual-specific tyrosine phosphorylation-regulated kinase 1A (DYRK1A). Consistently, a study showed that the level of DYRK1A in AD patients’ brains was higher compared to healthy people, and it was related to the occurrence and progression of AD. Urolithin A significantly decreased the activity of DYRK1A, which led to de-phosphorylation of tau by inhibiting the production of inflammatory cytokines caused by Aβ in AD-like mouse models [267]. Finally, recent research indicates that perturbated metabolic pathways like arachidonic acid (ARA), docosahexaenoic acid (DHA), eicosapentaenoic acid (EPA), L-phenylalanine, and cortisone are related to cognitive impairment after Aβ1-42 stimulation and hold promise as potential AD biomarkers [268].

#### 4.7.2. Clinical Studies

Perturbations in tyrosine and phenylalanine metabolic pathways can affect neurotransmitter biosynthesis and are closely associated with the progression of AD and PD in humans. Human metabolomics analysis reported altered 3-chlorotyrosine and L-tyrosine metabolites in saliva which exhibited high correlations with AD severity progression and concomitant depletion in vitamin B12 metabolism (Figure 3) [269]. A study conducted by Xu et al. reported metabolic sex-specific differences associated with plasma metabolites and AD endophenotypes. Specifically, increased levels of vanillylmandelate (tyrosine-dopamine pathway) and decreased levels of tryptophan betaine were associated with AD in females, whereas increased levels of KA were associated with AD in males [270].

### 4.8. Tyrosine Metabolites in PD

Tyrosine metabolism directly regulates the synthesis of L-3,4-dihydroxyphenylalanine (L-DOPA) urolithins into dopamine and, indirectly, the synthesis of norepinephrine [271]. At the same time, alterations in phenylalanine metabolism can cause changes in the levels of catecholamine neurotransmitters [272]. The enzyme tyrosine hydroxylase (TH) catalyzes the hydroxylation of L-tyrosine to L-DOPA, a crucial and rate-limiting step for the synthesis of dopamine (DA), norepinephrine (NE), and epinephrine (EP). Indeed, an alteration of the presynaptic dopamine machinery has been documented, evidenced by the progressive increase in homovanillic acid levels in the cerebrospinal fluid, which correlates with motor deterioration in Parkinson’s patients [273]. Of importance, emerging evidence reports that neuronutrients can modulate L-DOPA synthesis through hydroxylation of L-tyrosine for the treatment of PD. In beetroot and tomato fruit, the genes CYP76AD1 and CYP76AD6, which catalyze the hydroxylation of tyrosine to L-DOPA, have been identified [274]. Consistent with the brain resilience potential of natural sources, the tomato plant has been shown to have higher levels of most amino acids in fruits that accumulate L-DOPA. These include alanine, asparagine, glutamine, glycine, isoleucine, leucine, lysine, phenylalanine, methionine, serine, threonine, valine, and tryptophan, and are associated with greater antioxidant defense capacity compared to synthetic L-DOPA in the treatment of PD [274]. Furthermore, bioactive components of palm fruit increased tyrosine hydroxylase levels in the brain, stimulating the growth of colonic microbiota [275]. Another example is sodium benzoate, a metabolite of cinnamon, which increased neuroprotective molecules and protected dopaminergic neurons in a mouse model of PD. Specifically, sodium benzoate increased TH protein expression to produce dopamine and improved motor activity in the striatum of old C57/BL6 mice and in astrocytes thanks to rapid CREB activation and transcription of GDNF [276]. Finally, the dose of the neuronutrient sesamin (25 and 50 mg/kg) promoted a neuroprotective action against habit learning and spatial memory deficits by activating the dopamine neuronal system via its modulatory effects on the NMDAR-ERK1/2-CREB system in patients with PD receiving L-DOPA [277]. Overall, identifying specific tyrosine metabolites that are regulated by neuronutrients may provide new preventive and therapeutic targets. By understanding how neuronutrients influence these metabolites, it may be possible to activate the Nrf2 pathway and specific brain resilience genes and proteins, potentially alleviating PD symptoms.

**Table 2 ijms-26-09391-t002:** Effects of metabolites in molecular pathways involved in AD, PD, and autism.

Metabolites	Upregulation	Downregulation	Outcomes: AD	Outcomes: PD	Outcomes: Autism	Ref.
Kynurenic acid	Erk/JNK/MAPK	DR3/IKK/NF-κB	Neuroprotective effects	Protection neuronal	Neuroprotection	[185]
SCFAs	Nrf2 PI3K/Akt/mTOR	IL-6, IL-8, Il-12, IL-17, IL-1β, TNF-α	Strengthen BBB integrity	Neuroprotective role	Reduction in autism severity	[197,226]
Dihydroasparagusic acid	-	TNF-α, PGE2 Cyclooxygenase-2 Lipoxygenase	Inhibits the processes neuroinflammatory	Inhibits oxidative processes	-	[239]
S-Allyl cysteine	Nrf2/TLR4	-	Neuroprotective and anti-amyloidogenic effects	Neuroprotective effects	-	[241]
6-(Methylsulfinyl) hexyl isothiocyanate	Nrf2	GSK-3β/NF-κB	Protection from oxidative stress and inflammation	Preserved nigral dopaminergic neurons	-	[251]
Hydrogen sulfide	Akt/glycogen synthase kinase-3β/β-catenin	-	Increases neurogenesis improved cognitive deficits	Neuroprotective properties	-	[256]
Tyrosine	PI3K/AKT/Nrf2	MAPK/NF-κB	Neuroprotective effects	Protective action on dopaminergic neurons	Alleviates behavioral disorders, social communication deficits and reduced repetitive behavior	[262,278]

### 4.9. Tyrosine Metabolites in Autism

#### 4.9.1. Preclinical Studies

Recent preclinical studies showed that L-tyrosine supplementation significantly mitigated ASD-like behavioral disorders, alleviated social communication deficits, and reduced repetitive behavior in autistic mice. L-tyrosine also regulated colonic barrier damage and the gut microbial composition and function strongly connected to the hippocampal genes and neurotransmitters affected by L-tyrosine. Microbiota transplantation from L-tyrosine-treated mice maintained physiological L-tyrosine levels and prevented behavioral deficits associated with ASD [278]. Moreover, an essential neuronutrient known as sialic acid can attenuate behavioral deficits in autistic rats. Specifically, metabolomic analysis identified differentially abundant metabolites (i.e., L-tyrosine, tyrosine, phenylacetylglycine, biopterin, bilirubin, thiamine, and biotin) regulated by sialic acid supplementation to ameliorate ASD-like phenotypes such as social behavior and learning and memory impairment, thus supporting its potential therapeutic role in ASD [279].

#### 4.9.2. Clinical Studies

Phenylalanine and tyrosine are metabolized by intestinal bacteria, resulting in the formation of 4-ethylphenol and p-cresol. These compounds are then conjugated by the host, resulting in 4-ethylphenyl sulfate and p-cresyl sulfate, respectively. Both compounds derive predominantly from bacterial fermentation of phenylalanine, with a minor contribution from tyrosine [280]. However, the presence of high amounts of 4-hydroxyphenylpyruvic acid, phenylpyruvic acid, and phenylacetic acid, which are derived exclusively from the catabolism of phenylalanine, supports the existence of a greater concentration of phenylalanine in the urine of autistic patients compared to their typically developing siblings. Accordingly, recent human studies have shown that a subset of ASD patients display increased circulation levels of the tyrosine metabolite, p-cresol sulfate, related to dysbiosis of the intestinal microbiota [280]. Importantly, p-cresol has the ability to cross the BBB via the gut–brain axis [280]. In particular, abnormal presence of intestinal Clostridium sp. has been linked to high levels of p-cresol in ASD children [280]. It is noteworthy that p-cresol and derivatives are microbial metabolites produced by bacterial fermentation of phenylalanine and tyrosine in the colon. These metabolites display potentially detrimental effects in the gut microbiota [280]. Consistently, evidence has shown that elevated levels of p-cresol and/or p-cresol sulfate in the urine of children vs. controls contribute to ASD core symptoms. Importantly, p-cresol exhibits antibiotic effects, destroying beneficial bacteria of the gut microbiota, causing dysbiosis [280]. Furthermore, p-cresol impairs ATP production and subsequently increases oxidative stress in mitochondria. In the brain, however, p-cresol has been recently shown to impair neuronal development and inhibit dopamine beta-hydroxylase, which in turn converts dopamine to noradrenaline. Intriguingly, microbiota transplant therapy effectively reduced p-cresol sulfate levels to normal, leading to improvements in ASD symptoms [281]. Furthermore, increased urinary excretion of another metabolite m-tyrosine (3-hydroxyphenylalanine), a tyrosine analog, reduced brain catecholamines and caused symptoms of autism (stereotypical behavior, hyperactivity, and hyper-reactivity) in patients [282]. Lastly, aberrant metabolic profiles in the phenylalanine, tyrosine, and tryptophan pathways have been also documented.

Disturbances in these metabolic pathways primarily lead to increased phenylalanine levels and decreased tyrosine levels. This imbalance also causes increased concentrations of bacterial degradation products, including phenylpyruvic acid, phenylacetic acid, and 4-ethylphenyl sulfate as observed in the urine of children with ASD [283].

## 5. Neuro-Epigenetic Interactions: Role of Nrf2 in Nervous System Disorders

The recent literature demonstrated that the transcriptional expression of Nrf2 and brain resilience genes is thinly controlled by neuroepigenetic interactions [284]. Epigenetic mechanisms, such as histone modifications, DNA methylation, and gene silencing mediated by microRNAs (miRNAs) provide an adaptive level of control over gene expression, allowing the organism to adapt to a variable environment [285]. Specifically, DNMTs transfer (within CpG dinucleotides) a methyl group to the 5′ position of the cytosine residue, catalyzing DNA methylation in the brain [285]. Several miRNAs have been found to regulate the Nrf2-Keap1 pathway [285]. miRNAs, transcribed from genetic loci, are small (20–22 nucleotides) non-protein-coding RNAs [286]. miRNAs regulate gene expression by inhibiting translation or inducing degradation of their target mRNAs [286]. The regulation of gene expression is a finely tuned process that defines cellular identity and is closely associated with both brain health and/or disease [287]. Neuroepigenetic regulation is essential in the processes of neuronal development, differentiation, and synaptic plasticity for normal brain function. Aberrant neuroepigenetic changes play a major role in numerous pathologies, including NDs and ASD [26,27]. Indeed, miR-34c and miR-124-3p targeting SP1 mRNA have shown to exert a neuroprotective role via attenuating the hyperphosphorylation of tau-induced cell apoptosis in AD [287]. Biological processes can be modulated through epigenetic modifications, or rather non-genetic pathways involving redox resilience signaling [288]. Nrf2, encoded by the NFE2L2 gene, is a redox-sensitive transcriptional factor widely recognized as a master regulator of the brain resilience response. Its function is to modulate the expression of a large number of antioxidant and cytoprotective genes [289]. In particular, Nrf2 expression is highly controlled. Under normal cellular environment, Nrf2 levels are low due to the action of an E3 ubiquitin ligase complex containing the substrate adaptor protein, Kelch-like ECH-associated protein 1 (Keap1) [3]. At this point, Keap1 binds to Nrf2 and promotes its degradation through the ubiquitin–proteasome pathway. However, in the presence of excessive stress, Nrf2 signaling is induced through modifications of key cysteine residues in keap1, which leads to conformational changes and prevents Nrf2 degradation. This allows the accumulation of newly synthesized Nrf2, which can translocate into the nucleus, bind to the ARE (antioxidant response element) sequence in the promoter regions of Nrf2-stress resilience genes, and recruit the transcriptional machinery [1,289,290,291,292]. The rise in Nrf2 expression can also be controlled at the epigenetic level. Recently, several studies have demonstrated that Nrf2 directly influences epigenetic mechanisms by regulating the expression of DNMTs, HDACs, and miRNAs [293]. For instance, hypermethylation of the first five CpG sites in the Nrf2 gene promoter represses its protective action, leading to prostate tumorigenesis [294]. In contrast, the frequency of Nrf2 gene promoter demethylation was significantly higher in colorectal cancer [295]. Epigenetically, Nrf2 is considered a promising therapeutic target to prevent the development and progression of cancer [295], cardiovascular diseases [296], and neurodegeneration [297]. Indeed, for the first time, ARE sequences in the promoter regions of genes encoding the epigenetic factors HDAC, DNMT, and proteins involved in miRNA biogenesis have been identified [298]. This suggests that Nrf2 can also influence the expression of these genes after binding to their promoters [298]. Nrf2 modulates the levels of specific miRNAs including miR-365-1/miR-193b cluster, miR-29-b1, miR-125-b1, and miR-155-5p through targeted miRNA degradation (TDMD) [293,299]. This research significantly elucidates other functionalities of Nrf2, identifying an epigenetic regulatory role complementing its traditional antioxidant function. This important discovery extends the comprehension of Nrf2 regulatory mechanisms, which should be taken into account in developing new targeted therapeutic approaches in various diseases [293]. The cellular availability of the ARE for gene activation by Nrf2 is finely regulated by the competitive binding of BACH1 and BACH2 to the same sequence motif. The correct interaction between Nrf2, BACH1, and BACH2 is essential for activating the resilience response to stress and for the expression of antioxidant genes that ensure cytoprotection [300]. Gene activation by Nrf2 is also associated with the recruitment of histone acetyltransferases (HATs), which increase local acetylation of histones. This acetylation makes chromatin more accessible for the binding of transcription factors, thus facilitating the activation of gene transcription. Similarly, an NAD+-dependent histone deacetylase, sirtuin 6, a member of the sirtuin family, for deacetylating and stabilizing Nrf2, increases the expression of antioxidant genes against ROS overload. On the other hand, pharmacological inhibition of Nrf2 removed this beneficial modulatory effect of Sirt6 on microglial activity [301]. In neurons, Nrf2 repression induced by epigenetic silencing at the NFE2L2 promoter is accompanied by histone H3 hypoacetylation [302]. Interestingly, hypo-expression of Nrf2 by epigenetic repression during neuronal differentiation and circuit formation leads to impaired intrinsic antioxidant genes Hmox1, Srxn1, xCt, Cat, and Gclc, which cannot be reversed by HDAC inhibition in very young neurons, suggesting an intrinsic neurodevelopmental role. Indeed, recent research hypothesized that Nrf2 repression is required for proper neurodevelopment due to its interference with Wnt signaling activation, which is essential for various brain developmental processes including neurite outgrowth and synapse formation [302].

### 5.1. Nrf2 Epigenetic Regulation in AD and PD

Depending on the pathology and cell type, Nrf2 signaling needs to be activated (as in neurodegenerative diseases and aging) or repressed (as in cancer). For example, neurons exhibit deficient Nrf2 signaling, whereas glial cells and their derived brain tumors, such as glioblastomas, exhibit high Nrf2 activity. This high activity is associated with a poor prognosis, suggesting a crucial role in tumor progression and treatment resistance. The fact that the NFE2L2 gene is widely expressed in glial cells and astrocytes supports the hypothesis that neurons depend on these non-neuronal cells for their antioxidant defense [303]. Neuronal activation of Nrf2 signaling through dimethyl fumarate promotes brain resilience effects during AD and PD both in human and mouse models [304,305]. Mapping ARE-bearing enhancers targeted by Nrf2 and/or BACH1 in the several brain cell types and understanding the contribution of resilience target genes to different nervous system disorders could enable more precise approaches for targeting Nrf2 signaling and promote neuroprotection. Indeed, a recent analysis revealed that BACH1 target genes involved in several neuroprotective pathways, including those related to stress resilience response, are repressed in the ventral midbrain during neurodegeneration [306]. Studies using a mouse model of PD demonstrated that genetic deletion or pharmacological inhibition of BACH1 can promote an Nrf2-mediated increase in resilience target genes and protect dopaminergic neurons from degeneration [306]. In the AD brain, alterations in global DNA hypermethylation levels have been reported by several studies, but results are different among them [307]. Finally, a study reported that inhibition of the G9a/GLP complex by UNC0642 increased Nrf2 and Heme oxygenase decycling 1 (Hmox1) gene expression, reducing DNA-methylation and di-methylation of lysine 9 of histone H3 and increasing hydroxymethylation in the hippocampus, ultimately promoting neuroprotective effects in AD mice [308].

### 5.2. Nutritional Modulators Regulate Epigenetic Modifications Targeting the Nrf2 Pathway

Epigenetic modulation of the Nrf2 pathway by nutritional compounds has gained interest in recent years [309]. Nutrients, particularly polyphenols, can regulate epigenetic changes in Nrf2 by modulating various signaling pathways through its activation or inactivation [309,310]. Abnormal methylation patterns in the Nrf2 gene promoter region have also been observed to be implicated in disease progression [310]. In this regard, numerous neuroactive nutrients can strengthen brain resilience by positively regulating the key signaling pathways involved in epigenetic modifications. These nutrients act through several kinases, including MAPK (such as ERK and JNK), AKT, PI3K, AMPK, and protein kinase C (PKC), which can phosphorylate threonine, tyrosine, and serine residues of Nrf2 [309,310,311,312,313]. At the same time, glycogen synthase kinase-3 beta (GSK-3β) performs the opposite function, acting as an inhibitor of Nrf2 [314,315]. Interestingly, neuronutrients can also regulate the expression of genes associated with amino acid metabolism, specifically tryptophan and tyrosine, and of their metabolites such as KYNA and dopamine. In this regard, a recent study showed that a Traditional Chinese medicine known as Shenqi granules enhanced the activation of Nrf2 and the upregulation of genes encoding for detoxification enzymes and proteins, some of which may play an important role in the metabolism of amino acids. The neuroprotective effects are likely due to modulation of tryptophan and tyrosine metabolism by interacting with the NFE2L2 gene [309]. Of note, baicalein induces post-translational modifications to upregulate the Nrf2 pathway independently of Keap1 expression [310]. Importantly, forebrain neurons exhibit reduced levels of Nrf2, due to epigenetic repression of the Nrf2 gene promoter early in development. Furthermore, the little Nrf2 present in these neurons is highly unstable. This suggests that the endogenous amount of Nrf2 may not be sufficient to adequately support the necessary antioxidant and proteostatic responses [316]. In fact, forebrain neurons are more vulnerable to oxidative stress and other cellular damage than astrocytes due to their limited capacity for Nrf2-mediated antioxidant defense [316]. Specifically, researchers have shown that activation of neuronal Nrf2 expression using gRNA-targeted deactivated Cas9-based transcriptional activation complexes is sufficient to trigger Nrf2-dependent resilience pathways that facilitate the clearance of α-synuclein aggregates [316]. Essentially, the idea of epigenetically reactivating Nrf2 in neurons through specific polyphenols and/or nutraceuticals may enhance their ability to handle oxidative stress and prevent the buildup of misfolded proteins like α-synuclein. In this way, Nrf2 modulators such as tert-butyl hydroquinone (tBHQ) and sulforaphane induce an ARE-mediated genetic response in astrocytes, which ultimately confers neuroprotection to vulnerable neurons [317]. Moreover, maternal resveratrol treatment for two months delayed neurodegeneration by modifying the methylation levels of CpG in Nrf2 and NF-kβ gene promoters in offspring through an upregulation of the AMPKα pathway with a concomitant increase in Pgc-1α gene expression, a downregulation of mTOR inhibition, and an increase in beclin-1 protein levels, thus reducing ROS levels and neuroinflammation [318]. In addition, low concentrations of sulforaphane (1.25 and 2.5 μM) upregulate the expression of Nrf2 and promote its nuclear translocation by reducing DNA methylation levels of the Nrf2 promoter, contributing to inhibit oxidative stress-induced AD pathogenesis [319]. Likewise, curcuma- and garlic-derived hybrids activate the Nrf2 signaling pathway and the ARE-regulated expression of its downstream target genes, such as HO-1 and NQO1, inducing epigenetic changes through modifications in miR-125b-5p expression [320]. Of note, pterostilbene (10 µM) reduces the activity of DNMTs and HDAC 1, 2, 3, and 4, facilitating Nrf2 promoter demethylation and the enhancement of antioxidant resilience genes and enzymes such as NQO1, HO-1, and SOD2 in cells [321]. Lastly, a moderate dose of 10 and 20 mg/kg of protocatechuic aldehyde significantly provides neuroprotection by attenuating dopaminergic neuronal loss via the activation of polo-like kinase 2 (PLK2)/pGSK3β/Nrf2 pathways in PD mice [322]. Overall, exploring the effects of neuronutrients on neuroepigenetic regulation could offer potential avenues for developing novel therapeutic strategies for NDs by targeting NFE2L2 gene and Nrf2-dependent brain resilience enzymes or epigenetically modifying protein biomarkers.

## 6. Conclusions

In summary, neuronutrients targeting the Nrf2 pathway and brain resilience genes could be neuronal-region- as well as cell-type-dependent. Future challenges include establishing novel targeted nutritional therapies to specifically enhance Nrf2 activation in particular cell types and/or brain regions, and to modulate the Nrf2 pathway in senescent cells. In this review, we discussed functional polyphenols such as ellagic acid, caffeoylquinic acid, fisetin, genistein, and anthocyanins, which act as effective Nrf2 inducers. Interestingly, we explored the Trp metabolic pathway and its major metabolites, including SCFA metabolites, sulfur-containing metabolites, and tyrosine metabolites, delving deeper into both the pathological and neuroprotective mechanisms related to CNS disorders. We noted that a personalized neuronutritional approach targeting the neuroprotective KYNA, which upregulates the Nrf2 pathway, represents a novel and promising strategy to promote brain resilience and shows high therapeutic potential to prevent and/or treat CNS diseases. Conversely, dysregulation of the KYNA/Nrf2 pathway and increased production of neurotoxic metabolites QUINA and IDO, which can cross the BBB and reach the brain, play a neuropathologic role mainly involving ROS overproduction and neuroinflammation. This ultimately leads to the onset and progression of AD, PD, and ASD. However, the role of epigenetic modifications in the pathogenesis of nervous system disorders remains underexplored. It is therefore necessary to better understand the roles and regulatory mechanisms of these crucial metabolites and their clinical applications. We believe that modulation of the Nrf2 signaling pathway by using specific neuropigenetic modifiers, such as polyphenols found in food products or dietary supplements, alone or in synergy with drugs, will help limit neuroinflammatory diseases, the aging process, and subsequently aging-related cognitive disorders and ASD pathogenesis, which are increasingly prevalent in the population. Recent evidence has shown that DNA demethylation promotes the Nrf2 cell signaling pathway, potentially enhancing the antioxidant system and brain resilience genes to counteract the development of AD [323]. Epigenetic or pharmacological inhibition of IDO and QUINA gene expression may be effective in suppressing AD and PD progression [162,167]. Additionally, we have verified that flavonoids can effectively modify Nrf2 signaling to prevent and/or treat various neuropathologies in vitro, in vivo, and in humans. Neuronutrients are receiving increasing attention for their crucial role as activators or inhibitors of Nrf2, depending on the dosage and the specific pathology. In this analysis, we highlighted how neuronutrients can delay the onset and/or prevent disease by acting in several ways: (1) by blocking oxidative stress and the pro-inflammatory cytokine cascade, through inhibition of Keap1 or activation of NFE2L2/Nrf2 gene expression and its downstream protein targets, such as HO-1, FoxO, GSH, SOD, and CAT; (2) by regulating Nrf2 signaling through various kinases, including PLK2/pGSK3β, ERK, PI3K/AKT, and MAPK; (3) by affecting epigenetic modifications (methylation/demethylation, acetylation/deacetylation of histones or miRNAs) in the NFE2L2 gene promoter region, which is also an important target of functional neuronutrients/polyphenols in preventing central lesions and related disorders. These findings underscore the importance of integrating epigenomic and metabolomic data to uncover the biological basis and potential biomarkers of neurodegeneration and neurodevelopmental disorders, thereby paving the way for novel personalized nutritional strategies for prevention, diagnosis, and treatment. However, future neuroscience research still needs to investigate the metabolomic, nutriepigenomic profiles for the targeted and personalized use of these compounds in public health strategies.

## Figures and Tables

**Figure 1 ijms-26-09391-f001:**
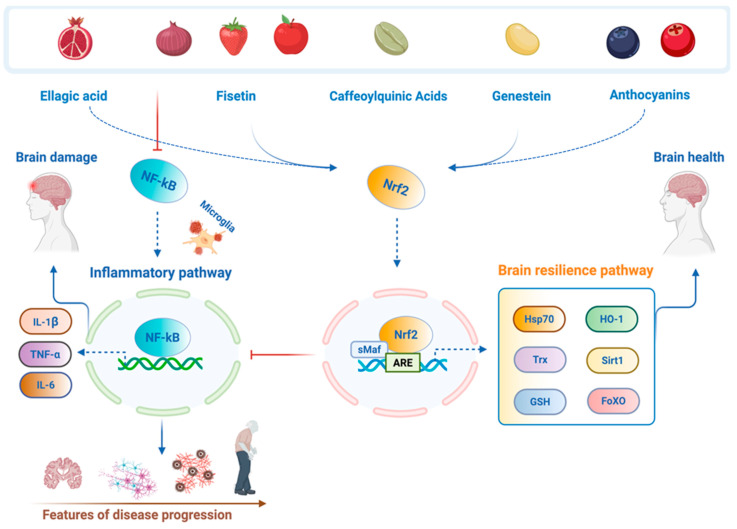
Neuronutrients modulate potential signaling pathways to improve brain resilience and health. Created in BioRender. Scuto, M. (2025) https://BioRender.com/oxdi21w.

**Figure 2 ijms-26-09391-f002:**
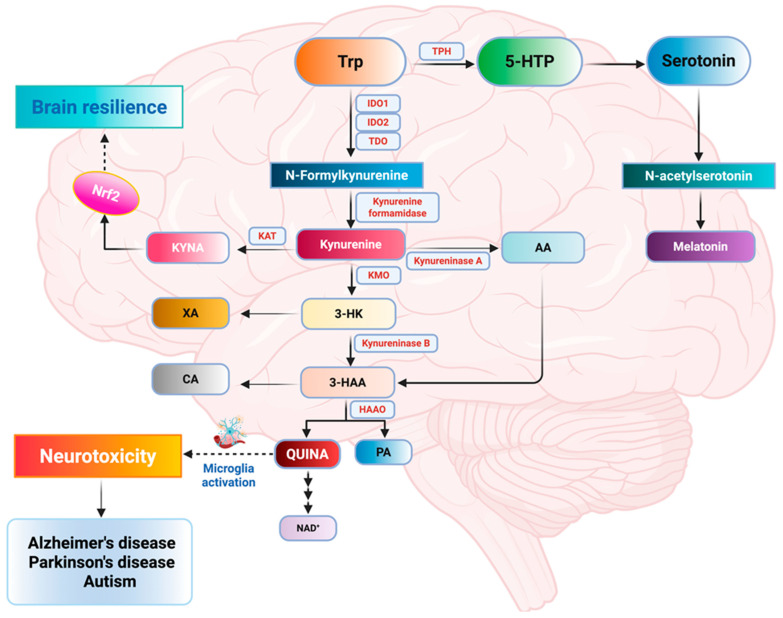
Overview of tryptophan metabolism and Nrf2 pathway in the brain. Created in BioRender. Scuto, M. (2025) https://BioRender.com/8y7pjn5. Tryptophan is a central metabolite capable of degradation through three specific pathways, shown in the schematic diagram, to generate different neuroactive metabolites. Abbreviations: TRP, tryptophan; IDO, indoleamine-2,3-dioxygenase; TDO, tryptophan-2,3-dioxygenase; KAT, kynurenine aminotransferase I–III; AA, anthranilic acid; 3-HK, 3-hydroxykynurenine; 3-HAA, 3-hydroxyanthrenillc acid; KMO, kynurenine 3-monooxygenase; HAAO, 3-hydroxyanthranilate 3,4-dioxygenase; KP, kynurenine pathway; KYNA, kynurenic acid; PA, picolinic acid; QUINA, quinolinic acid; TPH, tryptophan hydroxylase; CA, cinnabarinic acid; XA, xanthurenic acid; NAD+, nicotinamide adenine dinucleotide; 5-HTP, 5-hydroxytryptophan; TPH, tryptophan hydroxylase.

**Figure 3 ijms-26-09391-f003:**
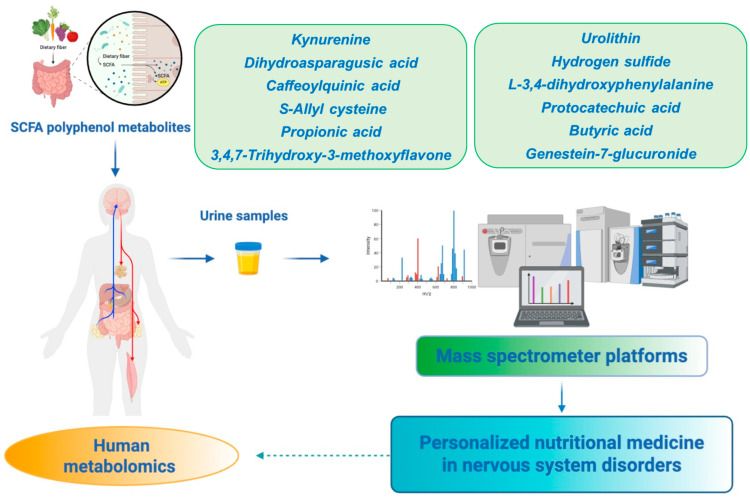
A schematic representation of nutri-metabolomics in nervous system disorders. Created in BioRender. Scuto, M. (2025) https://BioRender.com/84j68yl.

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
