# Peer review of "Neuronutrition and Nrf2 Brain Resilience Signaling: Epigenomics and Metabolomics for Personalized Medicine in Nervous System Disorders from Bench to Clinic"

_ijms, 2025, doi:10.3390/ijms26199391_

Round 1

Reviewer 1 Report

Comments and Suggestions for Authors

The manuscript entitled-Neuronutrition and Nrf2 brain resilience signaling: Epigenomics and Metabolomics for personalized medicine in nervous system disorders from bench to clinic- by Maria Concetta Scuto et al. provides a comprehensive review of literature that is highly relevant and timely, addressing important questions in the field. However, I have few suggestions that I believe will improve the quality of the manuscript. 

The title currently emphasizes Nrf2 brain resilience signaling, but the manuscript discusses a broad range of pathways, mechanisms, and neuronutrients, with Nrf2 being only one of several examples. I recommend adjusting the title to better reflect the content and balance of the review. For instance, the focus could be placed on neuronutrition, brain resilience, and personalized medicine, while mentioning Nrf2 as one of the key mechanisms rather than the central theme.

I appreciate the timely topic of the manuscript, particularly given the increasing prevalence of neurodegenerative, neurodevelopmental, and psychiatric disorders. However, several aspects could be clarified to improve readability and accessibility for a broader audience. First, key terms around which the review is structured, such as ‘brain resilience’ and ‘personalized neuronutrition’, should be clearly defined. The mechanisms by which the nervous system defends itself against harmful influences, and the distinction between ‘brain resilience’ and related concepts such as ‘neuroprotection’, should also be explicitly stated. Additionally, although the title emphasizes the NRF2 signaling pathway, this pathway is described in detail only at the end of the manuscript. I suggest restructuring the manuscript following a general-to-specific-to-individual format to better align the emphasis in the title with the content and to improve the logical flow of information.

Many sentences are overly long (3–4 lines) and contain multiple concepts. Breaking them down into shorter, focused statements would improve accessibility. Example: “From this new perspective, adopting a personalized neuronutritional approach…” is complex and could be split into two or three clearer sentences.

Comments on the Quality of English Language

The quality of English in the manuscript is good. However, there are several excessively long sentences, and I recommend shortening them or breaking them into multiple shorter sentences to improve readability.

Author Response

Thank you so much for taking the time to review this manuscript. Below, you will find detailed responses and any revisions/corrections in red, as well as in the submitted manuscript file.

The manuscript entitled-Neuronutrition and Nrf2 brain resilience signaling: Epigenomics and Metabolomics for personalized medicine in nervous system disorders from bench to clinic- by Maria Concetta Scuto et al. provides a comprehensive review of literature that is highly relevant and timely, addressing important questions in the field. However, I have few suggestions that I believe will improve the quality of the manuscript. 

The title currently emphasizes Nrf2 brain resilience signaling, but the manuscript discusses a broad range of pathways, mechanisms, and neuronutrients, with Nrf2 being only one of several examples. I recommend adjusting the title to better reflect the content and balance of the review. For instance, the focus could be placed on neuronutrition, brain resilience, and personalized medicine, while mentioning Nrf2 as one of the key mechanisms rather than the central theme.

Resp.: We have modified the title of paragraph 2, and we expanded the concepts on neuronutrition, brain resilience, and personalized medicine.

I appreciate the timely topic of the manuscript, particularly given the increasing prevalence of neurodegenerative, neurodevelopmental, and psychiatric disorders. However, several aspects could be clarified to improve readability and accessibility for a broader audience. First, key terms around which the review is structured, such as ‘brain resilience’ and ‘personalized neuronutrition’, should be clearly defined. The mechanisms by which the nervous system defends itself against harmful influences, and the distinction between ‘brain resilience’ and related concepts such as ‘neuroprotection’, should also be explicitly stated. Additionally, although the title emphasizes the NRF2 signaling pathway, this pathway is described in detail only at the end of the manuscript. I suggest restructuring the manuscript following a general-to-specific-to-individual format to better align the emphasis in the title with the content and to improve the logical flow of information.

Resp.: The aspects of "brain resilience" and "personalized neuronutrition" have been expanded and more clearly defined. The NRF2 signaling pathway has been moved to the beginning to improve the logical flow of information.

Many sentences are overly long (3–4 lines) and contain multiple concepts. Breaking them down into shorter, focused statements would improve accessibility. Example: “From this new perspective, adopting a personalized neuronutritional approach…” is complex and could be split into two or three clearer sentences.

The sentences of 3–4 lines have been modified.

Reviewer 2 Report

Comments and Suggestions for Authors

The authors summarized the neuronutrition in nervous system disorders from the epigenomics and metabolomics aspects. The topic is interesting, and the manuscript is overall comprehensive. A few points need to be addressed for further clarification.

1. Table 1. Line 3 neuronutrient was listed as blueberry instead of anthocyanins.

2. Figure 1. The arrows connecting different types of chemicals to pathways should be checked and reorganized in order to align with the text and table 1.

3 The section 2 focuses on discussing neuronutrition targeting brain resilience pathways. The introduction part emphasized the Nrf2 pathway, but the whole section also discussed and summarized the NF-κB too which lead more to section 3. Neuroinflammation: Role of Neuronutrients. For instance, the section 3 discussed genistein and fisetin, and neuronutrients inhibit neuroinflammatory cascade and promote brain resiliences. A portion of these topics was already mentioned in section 2. Section 2 categorized the molecular pathways of genistein included Nrf2/HO-1/PI3K, no NF-κB. However, in 3.1.1, it was described both upregulating Nrf2/HO-1 and downregulating NF-κB signaling pathways. Section 2 and 3 should be reorganized to clarify of the topics, and to avoid redundancy. The content should not conflict.

4. Figure 3. The compounds listed in the figure are the actual amino acids rather than their metabolites. This may be a bit confusing and better to label with their amino acid names.

5. It’s better to show the key structures from 4.6. Sulfur-containing nutrient metabolites structures in either figure 3 or a new figure. Same applies to ellagic acid, anthocyanins, caffeoylquinic acid, genistein and fisetin.

6. Page 34 Line 1422-1424. “We explored the Trp metabolic pathway and its major metabolites, including SCFA metabolites, sulfur-containing metabolites, and tyrosine metabolites, delving deeper into both the pathological and neuroprotective mechanisms related to CNS disorders.” SCFA metabolites, sulfur-containing metabolites, and tyrosine metabolites are not the metabolites of Trp.

7. Typo “genestein” should be correct in 3.1.1. Potential anti-neuroinflammatory effects of genestein.

Author Response

Thank you so much for taking the time to review this manuscript. Below, you will find detailed responses and any revisions/corrections in red, as well as in the submitted manuscript file.

The authors summarized the neuronutrition in nervous system disorders from the epigenomics and metabolomics aspects. The topic is interesting, and the manuscript is overall comprehensive. A few points need to be addressed for further clarification.

  1. Table 1. Line 3 neuronutrient was listed as blueberry instead of anthocyanins.

Resp.: done

  1. Figure 1. The arrows connecting different types of chemicals to pathways should be checked and reorganized in order to align with the text and table 1.

Resp.: we added the arrows connecting in Figure 1.

3 The section 2 focuses on discussing neuronutrition targeting brain resilience pathways. The introduction part emphasized the Nrf2 pathway, but the whole section also discussed and summarized the NF-κB too which lead more to section 3. Neuroinflammation: Role of Neuronutrients. For instance, the section 3 discussed genistein and fisetin, and neuronutrients inhibit neuroinflammatory cascade and promote brain resiliences. A portion of these topics was already mentioned in section 2. Section 2 categorized the molecular pathways of genistein included Nrf2/HO-1/PI3K, no NF-κB. However, in 3.1.1, it was described both upregulating Nrf2/HO-1 and downregulating NF-κB signaling pathways. Section 2 and 3 should be reorganized to clarify of the topics, and to avoid redundancy. The content should not conflict.

Resp.: We have improved information on the Nrf2 and NF-κB pathways in Sections 2 and 3 to eliminate overlap and to create a more linear discussion.

  1. Figure 3. The compounds listed in the figure are the actual amino acids rather than their metabolites. This may be a bit confusing and better to label with their amino acid names.

Resp.: We eliminated the amino acid structures to avoid confusion in Figure 3

  1. It’s better to show the key structures from 4.6. Sulfur-containing nutrient metabolites structures in either figure 3 or a new figure. Same applies to ellagic acid, anthocyanins, caffeoylquinic acid, genistein and fisetin.

Resp.: We modified Figure 3 and added nutrient metabolites

  1. Page 34 Line 1422-1424. “We explored the Trp metabolic pathway and its major metabolites, including SCFA metabolites, sulfur-containing metabolites, and tyrosine metabolites, delving deeper into both the pathological and neuroprotective mechanisms related to CNS disorders.” SCFA metabolites, sulfur-containing metabolites, and tyrosine metabolites are not the metabolites of Trp.

Resp.: We agree with the reviewer that SCFA metabolites, sulfur-containing metabolites, and tyrosine metabolites are not Trp metabolites; we have modified the title in the manuscript..

  1. Typo “genestein” should be correct in 3.1.1. Potential anti-neuroinflammatory effects of genestein.

Resp.: The word “genestein” has been corrected

Round 2

Reviewer 1 Report

Comments and Suggestions for Authors

The authors addressed all my concerns.